# Dual Instrumental Variable Regression

**Krikamol Muandet**
Max Planck Institute for Intelligent Systems
Tübingen, Germany
krikamol@tuebingen.mpg.de

**Arash Mehrjou**
Max Planck Institute for Intelligent Systems
ETH Zürich, Zürich, Switzerland
arash.mehrjou@inf.ethz.ch

**Si Kai Lee**
Booth School of Business
University of Chicago, USA
sikai.lee@chicagobooth.edu

**Anant Raj**
Max Planck Institute for Intelligent Systems
Tübingen, Germany
anant.raj@tuebingen.mpg.de

## Abstract

We present a novel algorithm for non-linear instrumental variable (IV) regression, DualIV, which simplifies traditional two-stage methods via a dual formulation. Inspired by problems in stochastic programming, we show that two-stage procedures for non-linear IV regression can be reformulated as a convex-concave saddle-point problem. Our formulation enables us to circumvent the first-stage regression which is a potential bottleneck in real-world applications. We develop a simple kernel-based algorithm with an analytic solution based on this formulation. Empirical results show that we are competitive to existing, more complicated algorithms for non-linear instrumental variable regression.

## 1 Introduction

Inferring causal relationships under the influence of unobserved confounders remains one of the most challenging problems in economics, health care, and social sciences [1, 2]. A typical example in economics is the study of returns from schooling [3], which attempts to measure the causal effect of education on labor market earnings. For each individual, the treatment variable $X$ represents the level of education and the outcome $Y$ represents how much they earn. However, one's level of education and income is likely confounded by the socioeconomic status or other unobserved confounding factors $H$ [1, Ch. 4].

Since randomized control trials are often infeasible in most economic studies, economists have turned to *instrumental variables* (IVs) or *instruments* derived from naturally occurring random experiments to overcome unobserved confounding. Informally, instrumental variables $Z$ are defined as variables that are associated with the treatment $X$, affect the outcome $Y$ only through $X$ and do not share common causes with $Y$. For instance, the season-of-birth was used as an instrument in [4] to estimate the impact of compulsory schooling on earnings. Because of the compulsory school attendance laws, an individual's season-of-birth, which is likely to be random, affects how long they actually remain in school, but not their earnings. Figure 1 illustrates this example. Finding valid instruments for specific problems is an essential task in econometrics [1] and epidemiology [5].

Although IV analysis is widely used, the statistical tools employed for estimating causal effect are fairly rudimentary. Most applications of instrumental variables utilise a two-stage procedure [1, 6–8]. For instance, the two-stage least squares (2SLS) relies on the assumption that the relationship between $X$ and $Y$ is linear [9]. It first estimates the conditional mean $\mathbb{E}[X|Z = z]$ via linear regression and then regresses $Y$ on the estimate of $\mathbb{E}[X|Z = z]$ to obtain an estimate of the causal effect. Since the first-stage estimate is by construction independent from confounders, the resultant

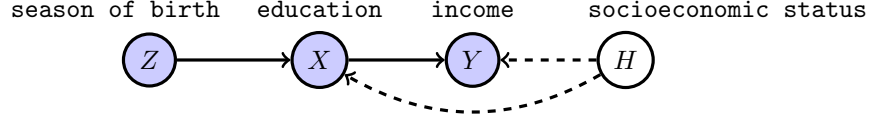

season of birth    education    income    socioeconomic status

Figure 1: A data generating process (DGP) with a hidden confounder $H$ and an instrument $Z$. A variation in $X$ comes from both $H$ and $Z$. Intuitively speaking, the external source of variation from $Z$ can help improve an estimation by removing the effect of $H$ on $X$.

causal estimate is therefore free from hidden confounding. In the non-linear setting, however, a poorly-fitted first-stage regression may result in inaccurate second-stage estimates [1, Ch. 4.6].

In this paper, we propose a novel procedure, DualIV, to directly estimate the structural causal function. Unlike previous works which extend 2SLS by employing non-linear models in place of their linear counterparts [7, 8], we solve the dual problem which can be expressed as a convex-concave saddle-point problem. Based on this framework, we develop a consistent reproducing kernel Hilbert spaces-based (RKHS) algorithm. Our formulation was inspired by the mathematical resemblance of non-linear IV to two-stage problems in stochastic programming [10–12].

The rest of the paper is organized in the following manner. Section 2 introduces the IV regression problem, reviews related work and identifies current limitations. We present our formulation in Section 3, followed by the kernelized estimation method in Section 4. Then, Section 5 reports empirical comparisons between DualIV and existing algorithms. Finally, we discuss the limitations of our procedure and suggest future directions in Section 6. All proofs can be found in Appendix E.

## 2   Instrumental variable regression

Let $X$, $Y$, and $Z$ be treatment, outcome, and instrumental variable(s) taking values in $\mathcal{X}$, $\mathcal{Y}$, and $\mathcal{Z}$, respectively. In this work, we assume that $\mathcal{Y} \in \mathbb{R}$, and $\mathcal{X}$ and $\mathcal{Z}$ are Polish spaces. We also assume that $Y$ is bounded, *i.e.*, $|Y| < M < \infty$ almost surely. Moreover, we denote unobserved confounder(s) by $H$. The underlying data generating process (DGP) is described by the causal graph in Figure 1 equipped with the following structural causal model (SCM):

$$Y = f(X) + \varepsilon, \quad \mathbb{E}[\varepsilon] = 0, \tag{1}$$

where $f$ is an unknown, potentially non-linear continuous function and $\varepsilon$ denotes the additive noise which depends on the hidden confounder(s) $H$. If $\mathbb{E}[\varepsilon \,|\, X] = 0$, we can estimate $f$ consistently from observational data via the standard least-square regression. This allows us to identify $\mathbb{E}[Y \,|\, \mathrm{do}(X = x)]$ where $\mathrm{do}(X = x)$ represents an intervention on $X$ where its value is set to $x$ [13].

For non-expert readers, we elaborate that $\mathrm{do}(X = x)$ here denotes a mathematical operator which simulates physical interventions by setting the value of $X$ to $x$, while keeping the rest of the model unchanged [14, Sec. 3.2.1]. That is, the conditional expectation $\mathbb{E}[Y \,|\, \mathrm{do}(X = x)]$ is computed with respect to the interventional distribution $\mathbb{P}(Y \,|\, \mathrm{do}(X = x))$. We can estimate $\mathbb{P}(Y \,|\, \mathrm{do}(X = x))$ if it is possible to directly manipulate $X$ and then observe the resulting outcome $Y$. In Figure 1, for instance, one may assign different levels of education to people and then observe their subsequent levels of income in the labor market. Unfortunately, such experiment is not always possible and we only have access to an observational distribution $\mathbb{P}(Y \,|\, X = x)$, which can be different from $\mathbb{P}(Y \,|\, \mathrm{do}(X = x))$. The discrepancy between interventional and observational distributions may result from the unobserved socioeconomic status, as illustrated in Figure 1.

When hidden confounders exist between $X$ and $Y$, the error term $\varepsilon$ in (1) is generally correlated with $X$. Hence, $\mathbb{E}[\varepsilon|X] \neq 0$ and it follows from (1) that

$$\mathbb{E}[Y \,|\, X = x] = f(x) + \mathbb{E}[\varepsilon \,|\, X = x], \tag{2}$$

which implies that $\mathbb{E}[Y \,|\, \mathrm{do}(X = x)] \neq \mathbb{E}[Y \,|\, X = x]$. Thus, standard least-square regression no longer provides a valid estimate of $f$ for making a prediction about the outcome of an intervention on $X$ [7, 8, 15]. To handle hidden confounders, we assume access to an instrumental variable(s) $Z$ which satisfies the following assumptions: (i) **Relevance:** $Z$ has a causal influence on $X$. (ii) **Exclusion restriction:** $Z$ affects $Y$ only through $X$, *i.e.*, $Y \perp\!\!\!\perp Z|X, \varepsilon$. (iii) **Unconfounded instrument(s):** $Z$ is independent of the error, *i.e.*, $\varepsilon \perp\!\!\!\perp Z$.

The properties of $Z$ imply that $\mathbb{E}[\varepsilon \,|\, Z] = 0$. Taking the expectation of (1) w.r.t. $Y$ conditioned on $Z$ yields the following integral equation

$$\mathbb{E}[Y|Z] = \int_{\mathcal{X}} f(x)\,\mathrm{d}\mathbb{P}(x|Z), \qquad (3)$$

which is a Fredholm integral equation of the first kind. Recent works in nonparametric IV regression have adopted this perspective [7, 8, 15, 16] despite the fact that solving (3) directly is an ill-posed problem as it involves inverting linear compact operators [15, 17, 18].

To illustrate the role of an instrument, we consider two special cases. When $X$ is perfectly correlated with $Z$, the treatment is uncorrelated with the hidden confounder. In other words, we recover the strong ignorability assumption [19, 20] required for causal inference. When $Z$ is independent of $X$, the instrument is useless as it has no predictive power over treatment so the structural function $f$ is unidentifiable from the data. Therefore, the most interesting cases lie between these two extremes, especially when $X$ and $Z$ are weakly correlated, see, *e.g.*, [21, 22][1, pp. 205–216].

## 2.1   Previous work

Early applications of instrumental variables often assume linear relationships between $Z$ and $X$ as well as $X$ and $Y$ [1, 23]. When there is a single endogenous variable and instrument, the structural parameter can be estimated consistently by the instrumental variable (IV) estimator [23]. Interestingly, we can obtain this estimate using a two-stage procedure: regress $X$ on $Z$ using ordinary least square (OLS) to calculate the predicted value of $X$ and used that as an explanatory variable in the structural equation to estimate the structural parameter using OLS. When there are multiple instruments, the two-stage least squares (2SLS) estimator is obtained by using all the instruments simultaneously in the first-stage regression. Wooldridge [24, Theorem 5.3] asserts that the 2SLS estimator is the most efficient IV estimator; see, *e.g.*, [1, 24] for a detailed exposition.

Recently, several extensions of 2SLS have been proposed to overcome the linearity constraint. The first line of work replaces linear regression by a linear projection onto a set of known basis functions [15, 16, 25, 26]. Chen and Christensen [27] provides a uniform convergence rate of this approach. However, there exists no principled way of choosing the appropriate set of basis functions. The second line of work replaces the first-stage regression by a conditional density estimate of $\mathbb{P}(X|Z)$ [28, 29]. Despite being more flexible, such approaches are known to suffer from the curse of dimensionality [30, Ch. 1]. Other extensions of 2SLS are DeepIV [8] and KernelIV [7] algorithms. In [8], (3) is solved by first estimating $\mathbb{P}(X|Z)$ with a mixture of deep generative models on which $f$ is learned using another deep neural network. Instead of neural networks, Singh et al. [7] proposes to model the first-stage regression using the conditional mean embedding of $\mathbb{P}(X|Z)$ [31–33] which is then used in the second-stage kernel ridge regression.

**The curse of two-stage methods.**   Two-stage procedures have two fundamental issues. First, such procedures violate Vapnik's principle [34]: "*[...] when solving a problem of interest, do not solve a more general problem as an intermediate step [...]*". Specifically, estimating the conditional density [8] or the conditional mean embedding [7] via regression in the first stage can be harder than estimating the parameter of interest in the second stage. The first stage is even referred as the "forbidden regression" in econometrics [1, Ch. 4.6]. On top of that, we usually only observe a single sample from each $\mathbb{P}(X|Z = z)$, which further increases the difficulty of the task. Second, although two-stage procedures are asymptotically consistent, the first-stage estimate creates a finite-sample bias in the second-stage estimate [1, Sec. 4.6.4]. This bias can be alleviated through sample splitting [35] which is also used in [7, 8]. Thus, two-stage procedures are less sample efficient and could yield biased estimates when run on the smaller datasets common in economics and social sciences.

The generalized method of moments (GMM) framework provides another set of popular approaches for estimating $f$ [36, 37]. Unlike two-stage procedures, GMM-based algorithms find a function $f$ that satisfies the orthogonality condition $\mathbb{E}[\varepsilon|Z] = 0$ directly. Specifically, if $g_1, g_2, \ldots, g_m$ are arbitrary real-valued functions, the orthogonality condition implies that $\mathbb{E}[(Y - f(X))g_j(Z)] = 0$ for $j = 1, \ldots, m$. The GMM estimate of $f$ can then be obtained by minimizing the quadratic form $\frac{1}{2}\sum_{j=1}^{m}\psi(f, g_j)^2$ where $\psi(f, g) := \mathbb{E}[(Y - f(X))g(Z)]$. This estimator can be interpreted as a generalization of the 2SLS estimator in the linear setting [6]. Recently, extensions of GMM-based methods where both $f$ and $g$ are parameterized by deep neural networks have successfully been

used to solve non-linear IV regression [38, 39]. In contrast, Muandet et al. [40] considers the set of RKHS functions which allow for an analytic formulation of the orthogonality condition.

## 3 Dual IV

In this section, we reformulate the integral equation (3) as an empirical risk minimization problem and present DualIV algorithm.

### 3.1 Empirical risk minimization

Let $\ell : \mathbb{R} \times \mathbb{R} \to \mathbb{R}_+$ be a proper, convex, and lower semi-continuous loss function for any value in its first argument.[1] Let $\mathcal{F}$ be an arbitrary class of continuous functions which we assume contains $f$ that fulfills the integral equation (3). Then, we can formulate (3) as

$$\min_{f \in \mathcal{F}} R(f) := \mathbb{E}_{YZ} \left[ \ell(Y, \mathbb{E}_{X|Z}[f(X)]) \right], \tag{4}$$

where $R(f)$ denotes the expected risk of $f$. To understand how (3) and (4) are related, let us consider the squared loss $\ell(y, y') = (y - y')^2$ and define $h(z) := \mathbb{E}_{X|z}[f(X)]$. Then, the solution to (4) is the minimum mean square error (MMSE) estimator $h^*(z) := \mathbb{E}[Y|z]$, which is exactly the LHS of (3). If there exists no $f \in \mathcal{F}$ for which $h^*(z) = \mathbb{E}[Y|z]$, we use $h^*(z)$ as the best MMSE approximation.

The key challenge here is that if $f$ is noncontinuous in $h(z)$, it is not assured to be consistently estimated even if $h(z)$ is estimated correctly [15]. We defer further discussion to Section 3.4. In addition, it remains cumbersome to solve (4) directly because of the inner expectation. To circumvent this, we look to similar two-stage problems in stochastic programming [10, 11]. For example, in [11], the problem of learning from conditional distributions was formulated in a similar fashion to (4). Moreover, [12] proposes the deconditional mean embedding (DME) which solves the integral equation (3) by performing a closed-form "inversion" of the conditional mean embedding of $\mathbb{P}(X|Z)$ (see [33, 42] for a review). By contrast, we solve the initial problem in (3) by resorting to the dual formulation of (4).

### 3.2 Dual formulation

To derive the dual form of (4), we employ two existing results, *interchangeability* and *Fenchel duality*, which we review; see, *e.g.*, [11, Lemma 1], [43, Ch. 14], and [10, Ch. 7] for more details.

**Theorem 1** (Interchangeability). *Let $\omega$ be a random variable on $\Omega$ and, for any $\omega \in \Omega$, the function $f(\cdot, \omega) : \mathbb{R} \to (-\infty, \infty)$ is proper and upper semi-continuous concave function. Then,*

$$\mathbb{E}_\omega \left[ \max_{u \in \mathbb{R}} f(u, \omega) \right] = \max_{u(\cdot) \in \mathcal{U}(\Omega)} \mathbb{E}_\omega[f(u(\omega), \omega)], \tag{5}$$

*where $\mathcal{U}(\Omega) := \{u(\cdot) : \Omega \to \mathbb{R}\}$ is the entire space of functions defined on the support $\Omega$.*

**Definition 2** (Fenchel duality). *Let $\ell : \mathbb{R} \times \mathbb{R} \to \mathbb{R}_+$ be a proper, convex, and lower semi-continuous loss function for any value in its first argument and $\ell_y^\star := \ell^\star(y, \cdot)$ a convex conjugate of $\ell_y := \ell(y, \cdot)$ which is also proper, convex, and lower semi-continuous w.r.t. the second argument. Then, $\ell_y(v) = \max_u \{uv - \ell_y^\star(u)\}$. The maximum is achieved at $v \in \partial \ell^\star(u)$, or equivalently $u \in \partial \ell(v)$.*

Applying the interchangeability and Fenchel duality to (4) yields the expected loss

$$
\begin{aligned}
R(f) &= \mathbb{E}_{YZ}[\max_{u \in \mathbb{R}} \{\mathbb{E}_{X|Z}[f(X)]u - \ell_Y^\star(u)\}] \\
&= \max_{u \in \mathcal{U}} \mathbb{E}_{YZ}[\mathbb{E}_{X|Z}[f(X)]u(Y, Z) - \ell_Y^\star(u(Y, Z))] \\
&= \max_{u \in \mathcal{U}} \mathbb{E}_{XYZ}[f(X)u(Y, Z)] - \mathbb{E}_{YZ}[\ell_Y^\star(u(Y, Z))]
\end{aligned}
$$

where $\mathcal{U}$ is the space of continuous functions over $\mathcal{Y} \times \mathcal{Z}$. Hence, (4) can be reformulated as

$$\min_{f \in \mathcal{F}} \max_{u \in \mathcal{U}} \mathbb{E}_{XYZ} \left[ f(X)u(Y, Z) \right] - \mathbb{E}_{YZ} \left[ \ell_Y^\star(u(Y, Z)) \right]. \tag{6}$$

Following [11], we will refer to $u \in \mathcal{U}$ as the *dual function*. Note that this function depends on only the outcome $Y$ and the instrument $Z$, but not the treatment $X$.

The advantages of our formulation (6) over (3) and (4) are twofold. First, there is no need to estimate $\mathbb{E}_{X|Z}[f(X)]$ or $\mathbb{P}(X|Z)$ explicitly. Second, the target function $f$ appears linearly in (6) which makes it convex in $f$. Since $\ell_y^\star$ is also convex, (6) is concave in the dual function $u$. Hence, (6) is essentially a convex-concave saddle-point problem for which efficient solvers exist [11].

For the squared loss $\ell(y, y') = (y - y')^2$, we have $\ell_y^\star(w) = wy + \frac{1}{2}w^2$ (see Appendix A for the derivation) and the saddle-point problem (6) reduces to

$$\min_{f \in \mathcal{F}} \max_{u \in \mathcal{U}} \ \Psi(f, u) := \mathbb{E}_{XYZ}\left[(f(X) - Y)u(Y, Z)\right] - \frac{1}{2}\mathbb{E}_{YZ}\left[u(Y, Z)^2\right]. \tag{7}$$

To solve (7), one can adopt an SGD-based algorithm developed by Dai et al. [11]. Alternatively, we propose in Section 4 a simple algorithm that can solve (7) in closed form.

### 3.3 Interpreting the dual function

The dual function $u(y, z)$ plays an important role in our framework. To understand its role, we consider the minimization and maximization problems in (7) separately. For any $f \in \mathcal{F}$, the maximization problem is $\max_{u \in \mathcal{U}} \mathbb{E}_{XYZ}[(f(X) - Y)u(Y, Z)] - \frac{1}{2}\|u(Y, Z)\|_{L^2(\mathbb{P}_{YZ})}^2$ where the first term can be viewed, loosely speaking, as a loss function and the second as a regularizer. Intuitively, we are seeking $u^* \in \mathcal{U}$ that is least orthogonal to the residual. Given $u^*$, the outer minimization problem $\min_{f \in \mathcal{F}} \mathbb{E}_{XYZ}\left[(f(X) - Y)u^*(Y, Z)\right]$ finds the function $f$ that yields the most orthogonal residual to $u^*$. Our procedure clearly differs from previous two-stage methods as the minimization and maximization stages are interdependent.

From the causal inference perspective, the residual contains the variation that cannot be explained by the current estimate of $f$ due to hidden confounding. We select $u$ that maximally reweights the residuals according to how inconsistent they are w.r.t. the unconfounded joint distribution of $Y$ and $Z$. Given $u$, we then select $f$ that minimizes the inconsistencies between the residuals and $u$. Hence, at the equilibrium, we are left with residuals uncorrelated with $Y$ and $Z$ which can be attributed to noise due to unobserved confounding.

Lastly, we draw a connection between (7) and GMM. Let $g_1, g_2, \ldots, g_m$ be real-valued functions on $\mathcal{Y} \times \mathcal{Z}$ and $\psi(f, g) := \mathbb{E}[(Y - f(X))g(Y, Z)]$. When $\mathcal{U} = \text{span}\{g_1, \ldots, g_m\}$, it is not difficult to show that $\max_{u \in \mathcal{U}} \Psi(f, u) = \frac{1}{2}\psi^\top \Lambda^{-1} \psi$ where $\Lambda := \mathbb{E}_{YZ}[\mathbf{g} \otimes \mathbf{g}]$ with $\mathbf{g} := (g_1(Y, Z), \ldots, g_m(Y, Z))^\top$; see Appendix B. That is, minimizing the above over $f$ yields a formulation that strongly resembles the GMM objective, with the dual function $u(Y, Z)$ playing a role similar to that of an instrument. However, we must clarify that $u$ cannot act as an instrument since it depends on $Y$ and thereby violates the exclusion restriction assumption. This ambiguity has been resolved in Liao et al. [44, Appendix F] by resorting to an alternative formulation similar to (4) and (7). Furthermore, we also note that AGMM [38] and DeepGMM [39] rely on minimax optimization, similar to (7), but were formulated based on the GMM framework.

### 3.4 Theoretical analysis

This section provides the conditions for which the true structural function $f^*$ can be identified by the optimum of the saddle-point problem (7). We lay out the assumptions needed for the optimal dual function $u^*$ to be unique and continuous, show that the saddle-point formulation (7) is equivalent to the problem (4) under the squared loss and prove that the solution of (7) given $u^*$ is indeed $f^*$.

**Assumption 1.** *(i) $\mathbb{P}(X|Z)$ is continuous in $Z$ for any values of $X$. (ii) The function class $\mathcal{F}$ is correctly specified,* i.e., *$f^* \in \mathcal{F}$.*

Following [11], we define the optimal dual function for any pair $(y, z) \in \mathcal{Y} \times \mathcal{Z}$ as $u^*(y, z) \in \arg\max_{u \in \mathbb{R}}\{\mathbb{E}_{X|z}[f(X) - y]u - (1/2)u^2\}$. Since this is an unconstrained quadratic program, $u^*(y, z)$ takes the form $\mathbb{E}_{X|z}[f(X)] - y$. Given Assumption 1 and the loss function $\ell$ is convex and continuously differentiable, it follows from [11, Proposition 1] that $u^*$ is unique and continuous.

Next, we shows that if $(f^*, u^*)$ is the saddle-point of (7), $f^*$ minimizes the original objective (4). The result follows from plugging $u^* = \mathbb{E}_{X|z}[f(X)] - y$ into the dual loss $\Psi(f, u)$ in (7); see Appendix E.1 for the detailed proof.

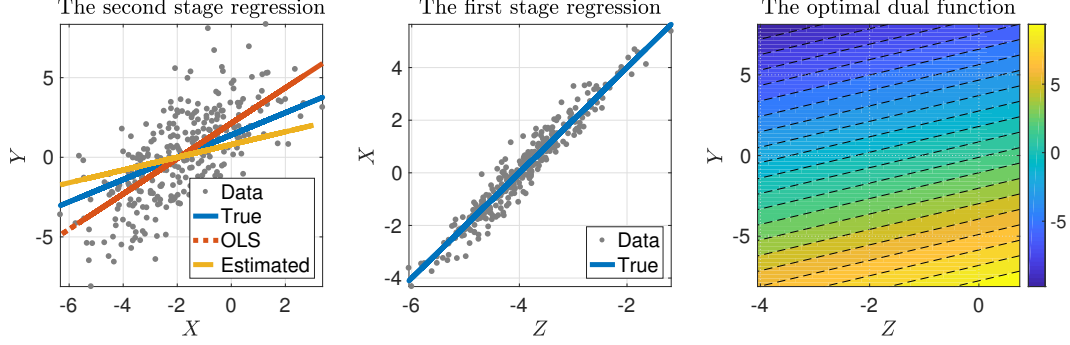

Figure 2: The dual function $u$ w.r.t. the current estimate $f$ in the linear setting (8). For each $y$ and $z$, $u$ can directly measures the discrepancy between $y$ and $\mathbb{E}_{X|z}[f(X)]$.

**Proposition 3.** *Let $\ell(y, y') = \frac{1}{2}(y - y')^2$. Then, for any fixed $f$, we have $R(f) = \max_u \Psi(f, u)$.*

By Proposition 3 and the convexity of the loss $\ell(y, y')$, we obtain the following result.

**Theorem 4.** *Let $\ell(y, y') = \frac{1}{2}(y - y')^2$ and assume that Assumption 1 holds. Then, $(f^*, u^*)$ is the saddle-point of a minimax problem $\min_{f \in \mathcal{F}} \max_{u \in \mathcal{U}} \Psi(f, u)$.*

By virtue of Theorem 4, we can identify the true function $f^*$ under relatively weak assumptions. In contrast, previous work usually require stronger assumptions such as the completeness condition [7, 15] which specifies that the first-stage conditional expectation $\mathbb{E}_{X|z}[f(X)]$ is injective, or $h(z) = \mathbb{E}_{X|z}[f(X)]$ is a smooth function of $z$ [7, 26, 27]. Since we do not perform first-stage regression, we only require $\mathbb{P}(X|Z)$ is continuous in $Z$ for any value of $X$. The assumption that (4) is correctly specified, *i.e.*, $f^* \in \mathcal{F}$, is standard in the literature [7, 15, 16].

As we can see, the optimal dual function $u^*(y, z) = \mathbb{E}_{X|z}[f(X)] - y$ acts as a residual function measuring the discrepancy between $y$ and $\mathbb{E}_{X|z}[f(X)]$ [11]. Remarkably, this makes it possible to approximate $R(f)$ in (4) without computing the expectation $\mathbb{E}_{X|z}[f(X)]$ explicitly. We will later exploit this property in selecting hyperparameters. Moreover, since $\mathbb{E}_{X|z}[f(X)]$ allows $X$ and $Z$ to have a non-linear relationship, $u$ can be non-linear even when the true structural function $f^*$ is linear. This flexibility enables $u$ to accommodate a larger class of functions that maps $Z$ to $X$. Figure 2 illustrates this given the following generative process:

$$Y = X\beta + e + \epsilon, \quad X = (1 - \rho)Z_1 + \rho e + \eta \tag{8}$$

where $e \sim \mathcal{N}(0, 2), Z_1 \sim \mathcal{N}(0, 2), \epsilon \sim \mathcal{N}(0, 0.1)$, and $\eta \sim \mathcal{N}(0, 0.1)$. The parameter $\rho$ controls the strength of the instrument w.r.t. hidden confounder $e$. Here, we set $n = 300, \beta = 0.7, \hat{\beta} = 0.4$, and $\rho = 0.2$ where $\hat{\beta}$ is an OLS estimate of $\beta$. Under this model, we have $u^*(y, z) \approx \hat{\beta}(1 - \rho)z - y$.

## 4 Kernelized DualIV

To demonstrate the effectiveness of our framework, we develop a simple kernel-based algorithm using the new formulation (7). To simplify notation, we denote by $W := (Y, Z)$ a random variable taking value in $\mathcal{W} := \mathcal{Y} \times \mathcal{Z}$. We pick $\mathcal{F}$ and $\mathcal{U}$ to be reproducing kernel Hilbert spaces (RKHSs) associated with positive definite kernels $k : \mathcal{X} \times \mathcal{X} \to \mathbb{R}$ and $l : \mathcal{W} \times \mathcal{W} \to \mathbb{R}$, respectively. Let $\phi : x \mapsto k(x, \cdot)$ and $\varphi : w \mapsto l(w, \cdot)$ be the canonical feature maps [45]. We assume both $\mathcal{F}$ and $\mathcal{U}$ are *universal* and hence are dense in the space of bounded continuous functions [46, Ch. 4].

Then, for any $f \in \mathcal{F}$ and $u \in \mathcal{U}$, we can rewrite the objective in (7) as

$$\Psi(f, u) = \mathbb{E}_{XW}[f(X)u(W)] - \mathbb{E}_{YZ}[Yu(Y, Z)] - \frac{1}{2}\mathbb{E}_W[u(W)^2]$$

$$= \langle \mathcal{C}_{WX}f - \mathbf{b}, u \rangle_{\mathcal{U}} - \frac{1}{2}\langle u, \mathcal{C}_W u \rangle_{\mathcal{U}}, \tag{9}$$

where $\mathbf{b} := \mathbb{E}_{YZ}[Y\varphi(Y, Z)] \in \mathcal{U}, \mathcal{C}_W := \mathbb{E}_W[\varphi(W) \otimes \varphi(W)] \in \mathcal{U} \otimes \mathcal{U}$ is a covariance operator, and $\mathcal{C}_{WX} := \mathbb{E}_{WX}[\varphi(W) \otimes \phi(X)] \in \mathcal{U} \otimes \mathcal{F}$ is a cross-covariance operator [47, 48] (see Appendix

---

**Algorithm 1** Kernelized DualIV

**Input:** Data $(x_i, y_i, z_i)_{i=1}^n$, kernel functions $k, l$, and a parameter grid $\Gamma$.
 1: Compute kernel matrices $\mathbf{K}_{ij} = k(x_i, x_j)$ and $\mathbf{L}_{ij} = l((y_i, z_i), (y_j, z_j))$.
 2: $(\lambda_1, \lambda_2) \leftarrow \texttt{SelectParams}(\mathbf{K}, \mathbf{L}, \Gamma)$.
 3: $\mathbf{M} \leftarrow \mathbf{K}(\mathbf{L} + n\lambda_1 I)^{-1}\mathbf{L}$.
 4: $\boldsymbol{\beta} \leftarrow (\mathbf{M}\mathbf{K} + n\lambda_2 \mathbf{K})^{-1}\mathbf{M}\mathbf{y}$.
**Output:** $f(x) = \sum_{i=1}^n \beta_i k(x_i, x)$.

---

C). Since (9) is quadratic in $u$, we have $\mathcal{C}_W u^* = \mathcal{C}_{WX} f - \mathbf{b}$. Substituting $u^*$ back into (9) yields

$$f^* = \arg\min_{f \in \mathcal{F}} \frac{1}{2} \left\langle \mathcal{C}_{WX} f - \mathbf{b}, \mathcal{C}_W^{-1}(\mathcal{C}_{WX} f - \mathbf{b}) \right\rangle_{\mathcal{U}} = (\mathcal{C}_{XW} \mathcal{C}_W^{-1} \mathcal{C}_{WX})^{-1} \mathcal{C}_{XW} \mathcal{C}_W^{-1} \mathbf{b}. \qquad (10)$$

We can view (10) as a generalized least squares solution in RKHS. Since $\mathcal{C}_W^{-1}$ and $(\mathcal{C}_{XW} \mathcal{C}_W^{-1} \mathcal{C}_{WX})^{-1}$ may not exist in general, we replace them with regularized versions $(\mathcal{C}_W + \lambda_1 \mathcal{I})^{-1}$ and $(\mathcal{C}_{XW} \mathcal{C}_W^{-1} \mathcal{C}_{WX} + \lambda_2 \mathcal{I})^{-1}$ where $\mathcal{I}$ is the identity operator and $\lambda_1, \lambda_2 > 0$ are regularization parameters.

Given an i.i.d. sample $(x_i, y_i, z_i)_{i=1}^n$ from $\mathbb{P}(X, Y, Z)$, we define $\Phi := [\phi(x_1), \dots, \phi(x_n)]$, $\Upsilon := [\varphi(y_1, z_1), \dots, \varphi(y_n, z_n)]$, and $\mathbf{y} := [y_1, \dots, y_n]^\top$. Then, we can estimate $\mathbf{b}$, $\mathcal{C}_W$, and $\mathcal{C}_{XW}$ with their empirical counterparts $\hat{\mathbf{b}} := n^{-1} \sum_{i=1}^n y_i \varphi(y_i, z_i) = n^{-1} \Upsilon \mathbf{y}$, $\widehat{\mathcal{C}}_{XW} := n^{-1} \sum_{i=1}^n \phi(x_i) \otimes \varphi(y_i, z_i) = n^{-1} \Phi \Upsilon^\top$ and $\widehat{\mathcal{C}}_W = n^{-1} \sum_{i=1}^n \varphi(y_i, z_i) \otimes \varphi(y_i, z_i) = n^{-1} \Upsilon \Upsilon^\top$. We denote the empirical version of (9) by $\widehat{\Psi}(f, u)$ and the estimate of $f^*$ by $\hat{f}$.

Next, we show that the representer theorem [49] for $\widehat{\Psi}(f, u)$ holds for both $f$ and $u$.

**Lemma 5.** *For any $f \in \mathcal{F}$ and $u \in \mathcal{U}$, there exist $f_{\boldsymbol{\beta}} = \sum_{i=1}^n \beta_i k(x_i, \cdot)$ and $u_{\boldsymbol{\alpha}} = \sum_{i=1}^n \alpha_i l(w_i, \cdot)$ for some $\boldsymbol{\alpha}, \boldsymbol{\beta} \in \mathbb{R}^n$ such that $\widehat{\Psi}(f, u) = \widehat{\Psi}(f_{\boldsymbol{\beta}}, u_{\boldsymbol{\alpha}})$.*

By virtue of Lemma 5, the solution to (10) can be expressed as $f(x) = \sum_{i=1}^n \beta_i k(x_i, x)$ where the coefficients $\boldsymbol{\beta}$ are given by the following proposition.

**Proposition 6.** *Given an i.i.d. sample $(x_i, y_i, z_i)_{i=1}^n$ from $\mathbb{P}(X, Y, Z)$, let $\mathbf{K} := \Phi^\top \Phi$ and $\mathbf{L} := \Upsilon^\top \Upsilon$ be the Gram matrices such that $\mathbf{K}_{ij} = k(x_i, x_j)$ and $\mathbf{L}_{ij} = l(w_i, w_j)$ where $w_i := (y_i, z_i)$. Then, $\hat{f} = \Phi \boldsymbol{\beta}$ where $\boldsymbol{\beta} = (\mathbf{M}\mathbf{K} + n\lambda_2 \mathbf{K})^{-1}\mathbf{M}\mathbf{y}$ and $\mathbf{M} := \mathbf{K}(\mathbf{L} + n\lambda_1 I)^{-1}\mathbf{L}$.*

Compared to previous work which involved conditional density estimation [8, 28, 29] and vector-valued regression [7] as first-stage regression, estimating the dual function $u$, a real-valued function, is arguably easier. This is especially so when $X$ and $Z$ are high-dimensional.

**Hyperparameter selection.** Our estimator depends on two hyper-parameters, $\lambda_1$ and $\lambda_2$. Given a dataset $(x_i, y_i, z_i)_{i=1}^{2n}$ of size $2n$, we provide a simple heuristic to determine the values of $(\lambda_1, \lambda_2)$. Ideally, if we know the optimal dual function $u^*$, we can interpret $u^*(y, z)^2$ as a loss function of $f^*$ at $(y, z)$, as discussed in Section 3.4. To this end, we first estimate $\hat{f}$ via Proposition 6 and $\hat{u}_\lambda := (\widehat{\mathcal{C}}_W + \lambda \mathcal{I})^{-1}(\widehat{\mathcal{C}}_{WX} \hat{f} - \hat{\mathbf{b}})$ on the first half of the data $(x_i, y_i, z_i)_{i=1}^n$. Next, the out-of-sample loss of $\hat{f}$ is evaluated on the second half $(x_i, y_i, z_i)_{i=n+1}^{2n}$ by

$$\widehat{R}(\hat{f}) = \frac{1}{n} \sum_{i=n+1}^{2n} (\mathbb{E}_{X|z_i}[\hat{f}(X)] - y_i)^2 \approx \frac{1}{n} \sum_{i=n+1}^{2n} \hat{u}_\lambda(y_i, z_i)^2. \qquad (11)$$

Note that $\hat{u}_\lambda = \Upsilon(\mathbf{L} + n\lambda I)^{-1}(\mathbf{K}\boldsymbol{\beta} - \mathbf{y}) = \Upsilon \boldsymbol{\alpha}$ where $\boldsymbol{\alpha} := (\mathbf{L} + n\lambda I)^{-1}(\mathbf{K}\boldsymbol{\beta} - \mathbf{y})$ and $\mathbf{K}, \mathbf{L} \in \mathbb{R}^{n \times n}$ are kernel matrices evaluated on $(x_i, y_i, z_i)_{i=1}^n$. Hence, $\widehat{R}(\hat{f}) \approx \boldsymbol{\alpha}^\top \widetilde{\mathbf{L}} \mathbf{1}/n$ where $\widetilde{\mathbf{L}}_{ij} = l((y_i, z_i), (y_j, z_j))$ for $i = 1, \dots, n$ and $j = n+1, \dots, 2n$ and $\mathbf{1}$ is the all-ones column vector. In practice, we fix $\lambda$ in $\hat{u}_\lambda$ to a small constant to stabilize the loss (11) and only optimize $(\lambda_1, \lambda_2)$ that appear in $\boldsymbol{\beta}$. Note that this procedure differs from the two-stage causal validation procedures used in [7, 8]. Alternatively, one may choose the hyperparameters by cross-validation with respect to (11).

Algorithm 1 outlines the kernelized DualIV method whose consistency is studied in Appendix D. We note that above algorithm involves matrix inversions ($\mathcal{O}(n^3)$) which become the primary computational bottlenecks when scaling to large datasets. To improve the scalability of our algorithm,

Table 1: Comparisons of IV regression methods in small (top) and medium (bottom) sample size regimes. We report the $\log_{10}$ mean squared error (MSE) and its standard deviations over 20 trials.

| $n = 50$ | $\text{Log}_{10}$ **Mean Squared Error (MSE)** | | | | |
|---|---|---|---|---|---|
| | $\rho = 0.1$ | $\rho = 0.25$ | $\rho = 0.5$ | $\rho = 0.75$ | $\rho = 0.9$ |
| 2SLS | $5.814 \pm 1.214$ | $6.013 \pm 0.827$ | $5.895 \pm 0.718$ | $5.625 \pm 1.182$ | $5.308 \pm 1.031$ |
| DeepIV | $5.127 \pm 0.043$ | $5.131 \pm 0.031$ | $5.133 \pm 0.072$ | $5.130 \pm 0.124$ | $5.127 \pm 0.061$ |
| KernelIV | $4.481 \pm 0.134$ | $4.460 \pm 0.095$ | $4.438 \pm 0.132$ | $4.433 \pm 0.100$ | $4.462 \pm 0.114$ |
| DeepGMM | $3.848 \pm 1.096$ | $2.899 \pm 1.638$ | $3.952 \pm 0.900$ | $4.148 \pm 0.556$ | $3.738 \pm 0.587$ |
| DualIV | $4.257 \pm 0.108$ | $4.210 \pm 0.126$ | $4.285 \pm 0.170$ | $4.286 \pm 0.126$ | $4.232 \pm 0.152$ |
| $n = 1000$ | | | | | |
| 2SLS | $8.236 \pm 0.117$ | $7.242 \pm 1.232$ | $8.290 \pm 1.132$ | $8.371 \pm 0.865$ | $8.544 \pm 1.109$ |
| DeepIV | $4.613 \pm 0.052$ | $4.618 \pm 0.048$ | $4.614 \pm 0.068$ | $4.701 \pm 0.040$ | $4.731 \pm 0.032$ |
| KernelIV | $4.189 \pm 0.046$ | $4.209 \pm 0.040$ | $4.199 \pm 0.043$ | $4.195 \pm 0.045$ | $4.194 \pm 0.055$ |
| DeepGMM | $4.090 \pm 0.691$ | $3.953 \pm 1.076$ | $4.392 \pm 0.561$ | $4.272 \pm 0.595$ | $4.415 \pm 0.522$ |
| DualIV | $4.143 \pm 0.117$ | $4.221 \pm 0.185$ | $4.104 \pm 0.102$ | $4.142 \pm 0.105$ | $4.127 \pm 0.106$ |

we can leverage the rich literature on large-scale kernel machines such as random Fourier features and Nyström method; see, e.g., Yang et al. [50] and references therein. Alternatively, we can employ stochastic gradient descent-based (SGD) algorithms similar to those proposed in Dai et al. [11, Algorithm 1] to solve the dual formulation (7) directly. This would also allow us to employ flexible models such as neural networks to parameterize the function classes $\mathcal{F}$ and $\mathcal{U}$. Recently, Liao et al. [44] has taken an important step in this direction and provided convergence analysis for neural networks under a similar formulation.

## 5 Experiments

In this section, we compare kernelized DualIV[2] with: (i) vanilla two-stage least squares (2SLS) [23], (ii) DeepIV [8], (iii) KernelIV [7] and (iv) DeepGMM [39]. To provide a fair comparison, we adhered to the provided hyperparameter settings. Given the low-dimensional nature of our experiments, we used DeepGMM's settings for low-dimensional scenarios in [39, Appendix B.2.1]. We ran 20 simulations of each algorithm for sample sizes of 50 and 1000 and calculated the $\log_{10}$ mean squared error and its standard deviations w.r.t. the true function $f$ for 2800 out-of-sample test points.

**Demand design.** We consider the same simulation as in [7, 8]: $Y = f(X) + \varepsilon$ where $Y$ is outcome, $X = (P, T, S)$ are inputs, and $Z = (C, T, S)$ are instruments. Specifically, $Y$ is sales, $P$ is price, which is endogenous, $C$ is a supply cost shifter (instrument), and $(T, S)$ are time of year and customer sentiment acting as exogenous variables. The aim is to estimate the demand function $f(p, t, s) = 100 + (10 + p)s\psi(t) - 2p$ where $\psi(t) = 2[(t - 5)^4/600 + \exp(-4(t - 5)^2) + t/10 - 2]$. Training data is sampled according to $S \sim \text{Unif}\{1, \ldots, 7\}$, $T \sim \text{Unif}[0, 10]$, $(C, V) \sim \mathcal{N}(\mathbf{0}, I_2)$, $\varepsilon \sim \mathcal{N}(\rho V, 1 - \rho^2)$, and $P = 25 + (C + 3)\psi(T) + V$. The parameter $\rho \in \{0.1, 0.25, 0.5, 0.75, 0.9\}$ controls the extent to which price $P$ is confounded with outcome $Y$ by supply-side market forces. In our notation, $X = (P, T, S)$, $Z = (C, T, S)$ and $W = (Y, Z) = (Y, C, T, S)$.

For DualIV, we used the Gaussian RBF kernel for both $k$ and $l$. In the experiments, the kernels on $X$ and $Z$ are product kernels, i.e., $k(x_i, x_j) = k_p(p_i, p_j)k_t(t_i, t_j)k_s(s_i, s_j)$ and $k(z_i, z_j) = k_c(c_i, c_j)k_t(t_i, t_j)k_s(s_i, s_j)$, and $l([y_i, z_i], [y_j, z_j]) = \exp([y_i - y_j, z_i - z_j]^\top V_{yz}^{-1}[y_i - y_j, z_i - z_j])$ where $V_{yz}$ is a symmetric bandwidth matrix. The values of all bandwidth parameters are determined via the median heuristic. We choose $(\lambda_1, \lambda_2)$ from $\{10^{-10}, 10^{-9}, \ldots, 10^{-1}\}$ via cross-validation. Once $(\lambda_1, \lambda_2)$ is chosen, we refit $\hat{f}$ on the entire training set.

Table 1 reports the results of different methods evaluated on the test data. First, we observe that 2SLS achieves the largest MSE in both regimes as expected because the linearity assumption is violated here. Second, in the small sample size regime, DeepIV achieves relatively larger MSE than the other non-linear methods. KernelIV, DeepGMM, and DualIV, on the other hand, have comparable performance, with DeepGMM having the lowest MSE. However, we note that the results attained

by DeepGMM were unstable out of the box and we had to reduce the variance of the initialization of the neural networks to 0.1 to obtain some degree of stability which is reflected in the standard deviations. We can fully attribute this variability to initialization as DeepGMM's default batch size of 1024 is larger than that of both training datasets so there is no sampling variability in the optimization process. This suggests that DeepGMM, like DeepIV, is relatively brittle compared to kernel-based methods in the small sample size regime. Furthermore, DeepGMM comes with an extensive hyperparameter selection process, which highlights its need for fine-tuning. Last but not least, DualIV is competitive to KernelIV across $\rho$ with slightly smaller MSE, which lends weight to our hypothesis that estimating the real-valued dual function is easier than vector-valued regression.

In the medium sample size regime, we observe that performance of DeepIV is in the same ballpark as the rest of the non-linear IV regression methods and the variance of DeepGMM is reduced, albeit still highest among the non-linear methods. The results of DualIV, KernelIV and DeepGMM are almost indistinguishable with DualIV having an edge as $\rho$ increases. This could mean accounting for both $Y$ and $Z$ is perhaps slightly more effective than $Z$ alone in the presence of greater confounding.

## 6   Conclusion

This paper proposes a general framework for non-linear IV regression called DualIV. Unlike previous work, DualIV does not require the first-stage regression which is the critical bottleneck of modern two-stage procedures. By exploiting tools in stochastic programming, we were able to reformulate the two-stage problem as the convex-concave saddle-point problem which is relatively simpler to solve. Instead of first-stage regression, DualIV requires the dual function $u(y, z)$ to be estimated, which is arguably easier than first-stage regression, especially when the instruments and treatments are high-dimensional. We demonstrate the validity of our framework with a kernel-based algorithm. Results show the competitiveness of our algorithm with respect to existing ones. Finally, potential directions for future work include (i) a minimax convergence analysis which could provide additional insight into the benefits of our framework, (ii) more flexible and scalable models such as deep neural networks as dual functions with stochastic gradient descent (SGD) [11], and (iii) applications to other two-stage problems in causal inference such as double ML [51].

### Broader impact

This work provides a new framework for non-linear instrumental variable regression which allows one to perform causal analysis under the presence of unobserved confounders. This could have a profound impact in other fields such as economics, social science, and epidemiology, among others. Understanding the role of instruments in the context of learning theory may also pave the way towards creating more robust and trustworthy machine learning algorithms that are capable of surviving in the world full of hidden biases.

### Acknowledgments and Disclosure of Funding

We are indebted to Rahul Singh and Arthur Gretton for their help with the KernelIV code which was used in our experiments. We thank Victor Chernozhukov, Elias Bareinboim, Sorawit Saengkyongam, Uri Shalit, Konrad Kording, Rahul Singh, Arthur Gretton, and You-Lin Chen for fruitful discussions as well as anonymous reviewers for the helpful feedback on our initial submission.

This work is funded by the federal and state governments of Germany through the Max Planck Society (MPG).

## Footnotes

[1]The function $f$ is *proper* if dom $f \neq \emptyset$ and $f(x) > -\infty$, $\forall x \in \mathcal{X}$. It is *lower (upper) semi-continuous* at $x_0 \in \mathcal{X}$ if for $\varepsilon > 0$ there exists a neighborhood $N(x_0)$ of $x_0$ such that $\varepsilon < (>)f(x) - f(x_0)$ for all $x \in N(x_0)$ [41].

[2]Our implementation is available at https://github.com/krikamol/DualIV-NeurIPS2020.

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
