[Supplementary Material]

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

 function for all $y \in \mathbb{R}$. It follows from the definition of Fenchel conjugate (see, *e.g.*, Definition 2 or [43, Ch. 14] and [10, Ch. 7]) that for any $y \in \mathbb{R}$,

$$\ell_y^\star(u) := \sup\left\{uv - \ell_y(v) \,:\, v \in \mathbb{R}\right\} = \sup\left\{uv - \frac{1}{2}(y-v)^2 \,:\, v \in \mathbb{R}\right\}. \tag{12}$$

Hence, $\ell_y^\star(u)$ is also a proper, concave, and upper semi-continuous function. Taking a derivative of $uv - \frac{1}{2}(y-v)^2$ w.r.t. $v$ and setting it to zero yield a critical point $v^* = u + y$ for any $u, y \in \mathbb{R}$. Since $uv - \frac{1}{2}(y-v)^2$ is a concave function in $v$, we can substituting $v^*$ back into (12) to obtain

$$\ell_y^\star(u) = u(u+y) - \frac{1}{2}(y - (u+y))^2 = u^2 + uy - \frac{1}{2}u^2 = uy + \frac{1}{2}u^2,$$

as required.

## B  Connection to generalized method of moments (GMM)

To understand the connection between DualIV and GMM, let us consider $\mathcal{U} := \operatorname{span}(g_1, \ldots, g_m)$ where $g_1, \ldots, g_m$ are arbitrary real-valued functions on $\mathcal{Y} \times \mathcal{Z}$. That is, for any $u \in \mathcal{U}$, we have $u = \sum_{j=1}^m \alpha_j g_j$ for some $(\alpha_1, \ldots, \alpha_m)^\top \in \mathbb{R}^m$. Then, we have

$$
\begin{aligned}
J(f) \;\; &:= \;\; \max_{u \in \mathcal{U}} \Psi(f, u) \\[2mm]
&= \;\; \max_{\boldsymbol{\alpha} \in \mathbb{R}^m} \mathbb{E}_{XYZ}\left[ (f(X) - Y)\left( \sum_{j=1}^m \alpha_j g_j(Y, Z) \right) \right] - \frac{1}{2}\mathbb{E}_{YZ}\left[ \left( \sum_{j=1}^m \alpha_j g_j(Y, Z) \right)^2 \right] \\[2mm]
&= \;\; \max_{\boldsymbol{\alpha} \in \mathbb{R}^m} \sum_{j=1}^m \alpha_j \mathbb{E}_{XYZ}\left[ (f(X) - Y)g_j(Y, Z) \right] - \frac{1}{2}\mathbb{E}_{YZ}\left[ \left( \sum_{j=1}^m \alpha_j g_j(Y, Z) \right)^2 \right] \\[2mm]
&= \;\; \max_{\boldsymbol{\alpha} \in \mathbb{R}^m} \boldsymbol{\alpha}^\top \boldsymbol{\psi} - \frac{1}{2}\boldsymbol{\alpha}^\top \Lambda \boldsymbol{\alpha},
\end{aligned}
$$

where we define $\boldsymbol{\alpha} := (\alpha_1, \ldots, \alpha_m)^\top$, $\boldsymbol{\psi} := (\psi(f, g_1), \ldots, \psi(f, g_m))^\top$, and $\Lambda := \mathbb{E}_{YZ}[\mathbf{g}(Y, Z) \otimes \mathbf{g}(Y, Z)]$ with $\mathbf{g}(Y, Z) := (g_1(Y, Z), \ldots, g_m(Y, Z))^\top$. Taking the derivative w.r.t. $\boldsymbol{\alpha}$ and setting it to zero yield

$$J(f) = \frac{1}{2}\boldsymbol{\psi}^\top \Lambda^{-1} \boldsymbol{\psi}. \tag{13}$$

In this case, the DualIV objective can be expressed in a quadratic form. In the language of GMM, $\boldsymbol{\psi}$ acts as a vector of moment conditions and $\Lambda$ acts as a weighting matrix [36, 37]. However, we have to reiterate that there is a fundamental difference here: the dual function $u \in \mathcal{U}$ cannot act as an instrument since it depends on $Y$ and thereby violates the exclusion restriction assumption. Recently, Liao et al. [44, Appendix F] provided a clarification on this connection by resorting to an alternative formulation similar to (4) and (7).

## C  Dual formulation in RKHS

In this section, we provide a detailed derivation of the dual formulation (9) when $\mathcal{F}$ and $\mathcal{U}$ are both reproducing kernel Hilbert spaces (RKHSs) associated with positive definite kernels $k : \mathcal{X} \times \mathcal{X} \to \mathbb{R}$ and $l : \mathcal{W} \times \mathcal{W} \to \mathbb{R}$. Let $\phi : x \mapsto k(x, \cdot)$ and $\varphi : w \mapsto l(w, \cdot)$ be the canonical feature maps of $k$ and $l$, respectively [45]. We assume throughout that both $\mathcal{F}$ and $\mathcal{U}$ are universal such that they are both dense in the space of bounded continuous functions (see, *e.g.*, [46, Ch. 4]). Furthermore, let $\mathrm{HS}(\mathcal{F}, \mathcal{U})$ be a Hilbert space of Hilbert-Schmidt operators mapping from $\mathcal{F}$ to $\mathcal{U}$ with an inner product $\langle \cdot, \cdot \rangle_{\mathrm{HS}}$ (see, *e.g.*, [33, Sec. 2.3]).

Then, for any $f \in \mathcal{F}$ and $u \in \mathcal{U}$, we can rewrite the objective in (7) as a functional

$$
\begin{aligned}
\Psi(f, u) &= \mathbb{E}_{XW}[f(X)u(W)] - \mathbb{E}_{YZ}[Yu(Y, Z)] - \frac{1}{2}\mathbb{E}_W[u(W)^2] \\
&= \mathbb{E}_{XW}[\langle f, \phi(X)\rangle_{\mathcal{F}}\langle u, \varphi(W)\rangle_{\mathcal{U}}] - \mathbb{E}_{YZ}[Y\langle u, \varphi(Y, Z)\rangle_{\mathcal{U}}] - \frac{1}{2}\mathbb{E}_W[\langle u, \varphi(W)\rangle_{\mathcal{U}}^2] \\
&= \mathbb{E}_{XW}[\langle f \otimes u, \phi(X) \otimes \varphi(W)\rangle_{\mathrm{HS}}] - \langle u, \mathbb{E}_{YZ}[Y\varphi(Y, Z)]\rangle_{\mathcal{U}} \\
&\quad - \frac{1}{2}\mathbb{E}_W[\langle u \otimes u, \varphi(W) \otimes \varphi(W)\rangle_{\mathrm{HS}}] \\
&= \langle \mathcal{C}_{XW}, f \otimes u\rangle_{\mathrm{HS}} - \langle u, \mathbf{b}\rangle_{\mathcal{U}} - \frac{1}{2}\langle \mathcal{C}_W, u \otimes u\rangle_{\mathrm{HS}} \\
&= \langle f, \mathcal{C}_{XW} u\rangle_{\mathcal{F}} - \langle u, \mathbf{b}\rangle_{\mathcal{U}} - \frac{1}{2}\langle u, \mathcal{C}_W u\rangle_{\mathcal{U}},
\end{aligned}
$$

where $\mathbf{b} := \mathbb{E}_{YZ}[Y\varphi(Y, Z)] \in \mathcal{U}$, $\mathcal{C}_W := \mathbb{E}_W[\varphi(W) \otimes \varphi(W)] \in \mathcal{U} \otimes \mathcal{U}$ is a covariance operator, and $\mathcal{C}_{XW} := \mathbb{E}_{XW}[\phi(X) \otimes \varphi(W)] \in \mathcal{F} \otimes \mathcal{U}$ is a cross-covariance operator [47, 48]. We used the reproducing property of $\mathcal{F}$ and $\mathcal{U}$ in the second equality. The third equality follows from the property of the rank-one operator, *i.e.*,

$$
\langle f, g\rangle_{\mathcal{F}}\langle v, u\rangle_{\mathcal{U}} = \langle f \otimes v, u \otimes g\rangle_{\mathrm{HS}(\mathcal{F}, \mathcal{U})}.
$$

We then used the definition of $\mathcal{C}_W, \mathcal{C}_{XW}$, and $\mathbf{b}$ to get the fourth equation. The last equation follows from the fact that $\langle L, f \otimes u\rangle_{\mathrm{HS}(\mathcal{U}, \mathcal{F})} = \langle f, Lu\rangle_{\mathcal{F}}$ for any Hilbert-Schmidt operator $L \in \mathrm{HS}(\mathcal{U}, \mathcal{F})$, $f \in \mathcal{F}$, and $u \in \mathcal{U}$.

## D Consistency

In this section, we show that the kernelized DualIV estimator is asymptotically consistent under the assumption that $\mathcal{C}_W^{-1}$ and $(\mathcal{C}_{XW}\mathcal{C}_W^{-1}\mathcal{C}_{WX})^{-1}$ exist. Under these assumptions, we show in (10) that the solution $f^*$ of the saddle-point problem (9) can be expressed as $f^* = (\mathcal{C}_{XW}\mathcal{C}_W^{-1}\mathcal{C}_{WX})^{-1}\mathcal{C}_{XW}\mathcal{C}_W^{-1}\mathbf{b}$. For simplicity, we assume further that the operator norm of the inverse covariance functions are bounded from below. The following theorem shows the consistency for the estimator $\hat{f}$ obtained via Proposition 6.

**Theorem 7.** *Let $\hat{f}_\lambda$ be an empirical estimator of $f^*$ obtained from Proposition 6 with the regularization parameters $\lambda := (\lambda_1, \lambda_2)$. Assume that $\mathcal{C}_W^{-1}$ and $(\mathcal{C}_{XW}\mathcal{C}_W^{-1}\mathcal{C}_{WX})^{-1}$ exist and the operator norm of the inverse are bounded. Then, for sufficiently slow decay of regularization parameters $\lambda_1$ and $\lambda_2$, $\hat{f}_\lambda$ is a consistent estimator of $f^*$ in RKHS norm, i.e., $\|\hat{f}_\lambda - f^*\|_{\mathcal{F}} \to 0$ as $n \to \infty$.*

The proof of this theorem can be found in Appendix E.4.

The critical drawback of Theorem 7 is that it assumes the existence of $\mathcal{C}_W^{-1}$ and $(\mathcal{C}_{XW}\mathcal{C}_W^{-1}\mathcal{C}_{WX})^{-1}$ which may not hold in general. Similar assumption was also made in Fukumizu et al. [52] who provided counterexamples of cases in which such an assumption does not hold; see, also, [31, 33, 52] for the detailed discussion. One potential direction for future work is thus to provide the consistency result of DualIV under relatively weaker assumptions.

## E Proofs

This section contains the detailed proofs.

### E.1 Proof of Proposition 3

**Proposition 3.** *Let $\ell(y, y') = \frac{1}{2}(y - y')^2$. Then, for any fixed $f$, we have $R(f) = \max_u \Psi(f, u)$.*

*Proof.* Taking $\ell$ in (4) to be $\frac{1}{2}(y - y')^2$, plugging $u^*(y, z) = \mathbb{E}_{X|z}[f(X)] - y$ into (7) yields

$$
\Psi(f, u^*) = \mathbb{E}_{XYZ}[(f(X) - Y)u^*(Y, Z)] - \frac{1}{2}\mathbb{E}_{YZ}[u^*(Y, Z)^2]
$$

$$= \mathbb{E}_{XYZ}[(f(X) - Y)(\mathbb{E}_{X|Z}[f(X)] - Y)] - \frac{1}{2}\mathbb{E}_{YZ}[(\mathbb{E}_{X|Z}[f(X)] - Y)^2]$$

$$= \mathbb{E}_{YZ}[(\mathbb{E}_{X|Z}[f(X)] - Y)(\mathbb{E}_{X|Z}[f(X)] - Y)] - \frac{1}{2}\mathbb{E}_{YZ}[(\mathbb{E}_{X|Z}[f(X)] - Y)^2]$$

$$= \mathbb{E}_{YZ}[(\mathbb{E}_{X|Z}[f(X)] - Y)^2] - \frac{1}{2}\mathbb{E}_{YZ}[(\mathbb{E}_{X|Z}[f(X)] - Y)^2]$$

$$= \frac{1}{2}\mathbb{E}_{YZ}[(\mathbb{E}_{X|Z}[f(X)] - Y)^2] = R(f),$$

as required. $\qquad\square$

### E.2 Proof of Lemma 5

**Lemma 5.** *For any $f \in \mathcal{F}$ and $u \in \mathcal{U}$, there exist $f_{\boldsymbol{\beta}} = \sum_{i=1}^{n} \beta_i k(x_i, \cdot)$ and $u_{\boldsymbol{\alpha}} = \sum_{i=1}^{n} \alpha_i l(w_i, \cdot)$ for some $\boldsymbol{\alpha}, \boldsymbol{\beta} \in \mathbb{R}^n$ such that $\widehat{\Psi}(f, u) = \widehat{\Psi}(f_{\boldsymbol{\beta}}, u_{\boldsymbol{\alpha}})$.*

*Proof.* Given a fixed sample $(x_i, y_i, z_i)_{i=1}^n$ of size $n$, any RKHSes $\mathcal{F}$ and $\mathcal{U}$ can be decomposed as $\mathcal{F} = \mathcal{F}_{\boldsymbol{\beta}} \oplus \mathcal{F}_{\perp}$ and $\mathcal{U} = \mathcal{U}_{\boldsymbol{\alpha}} \oplus \mathcal{U}_{\perp}$ where $\mathcal{F}_{\boldsymbol{\beta}}$ and $\mathcal{U}_{\boldsymbol{\alpha}}$ are respectively subspaces consisting of functions of the following forms:

$$f_{\boldsymbol{\beta}} = \sum_{i=1}^{n} \beta_i k(x_i, \cdot), \quad u_{\boldsymbol{\alpha}} = \sum_{i=1}^{n} \alpha_i l(w_i, \cdot),$$

for some $\boldsymbol{\beta} \in \mathbb{R}^n$ and $\boldsymbol{\alpha} \in \mathbb{R}^n$. The orthogonal subspaces $\mathcal{F}_{\perp}$ and $\mathcal{U}_{\perp}$ consist of elements which are orthogonal to $\mathcal{F}_{\boldsymbol{\beta}}$ and $\mathcal{U}_{\boldsymbol{\alpha}}$, respectively, *i.e.*, for any $f_{\boldsymbol{\beta}} \in \mathcal{F}_{\boldsymbol{\beta}}, f_{\perp} \in \mathcal{F}_{\perp}, u_{\boldsymbol{\alpha}} \in \mathcal{U}_{\boldsymbol{\alpha}}, u_{\perp} \in \mathcal{U}_{\perp}$, we have $\langle f_{\boldsymbol{\beta}}, f_{\perp} \rangle_{\mathcal{F}} = 0$ and $\langle u_{\boldsymbol{\alpha}}, u_{\perp} \rangle_{\mathcal{U}} = 0$. Any elements $f \in \mathcal{F}$ and $u \in \mathcal{U}$ can thus be expressed as $f = f_{\boldsymbol{\beta}} + f_{\perp}$ and $u = u_{\boldsymbol{\alpha}} + u_{\perp}$ where $f_{\boldsymbol{\beta}} \in \mathcal{F}_{\boldsymbol{\beta}}, f_{\perp} \in \mathcal{F}_{\perp}, u_{\boldsymbol{\alpha}} \in \mathcal{U}_{\boldsymbol{\alpha}}$, and $u_{\perp} \in \mathcal{U}_{\perp}$.

Next, recall that

$$\widehat{\Psi}(f, u) = \langle f, \widehat{\mathcal{C}}_{XW} u \rangle_{\mathcal{F}} - \langle u, \hat{\mathbf{b}} \rangle_{\mathcal{U}} - \frac{1}{2}\langle u, \widehat{\mathcal{C}}_W u \rangle_{\mathcal{U}}$$

where $\widehat{\mathcal{C}}_{XW} = n^{-1}\Phi\Upsilon^{\top}$, $\widehat{\mathcal{C}}_W = n^{-1}\Upsilon\Upsilon^{\top}$, $\hat{\mathbf{b}} = n^{-1}\Upsilon\mathbf{y}$, $\Phi = [k(x_1, \cdot), \ldots, k(x_n, \cdot)]$, $\Upsilon = [l(w_1, \cdot), \ldots, l(w_n, \cdot)]$, and $\mathbf{y} = [y_1, \ldots, y_n]^{\top}$. Using the above decomposition, we have

$$\begin{aligned}
\widehat{\Psi}(f, u) &= \langle f, \widehat{\mathcal{C}}_{XW} u \rangle_{\mathcal{F}} - \langle u, \hat{\mathbf{b}} \rangle_{\mathcal{U}} - \frac{1}{2}\langle u, \widehat{\mathcal{C}}_W u \rangle_{\mathcal{U}} \\
&= \langle f_{\boldsymbol{\beta}} + f_{\perp}, \sum_{i=1}^{n} \beta_i' k(x_i, \cdot) \rangle_{\mathcal{F}} - \langle u, \hat{\mathbf{b}} \rangle_{\mathcal{U}} - \frac{1}{2}\langle u, \widehat{\mathcal{C}}_W u \rangle_{\mathcal{U}} \\
&= \langle f_{\boldsymbol{\beta}}, \sum_{i=1}^{n} \beta_i' k(x_i, \cdot) \rangle_{\mathcal{F}} - \langle u, \hat{\mathbf{b}} \rangle_{\mathcal{U}} - \frac{1}{2}\langle u, \widehat{\mathcal{C}}_W u \rangle_{\mathcal{U}},
\end{aligned}$$

where $\beta_i' := n^{-1}\langle l(w_i, \cdot), u \rangle_{\mathcal{U}}$. Since the choice of $u$ is arbitrary, the minimizer of $\widehat{\Psi}(f, u)$ with respect to $f$ lives in the subspace $\mathcal{F}_{\boldsymbol{\beta}}$.

Similarly, we can write $\widehat{\Psi}(f, u)$ for any $f \in \mathcal{F}$ as a function of $u \in \mathcal{U}$ as

$$\begin{aligned}
\widehat{\Psi}(f, u) &= \langle \widehat{\mathcal{C}}_{WX} f, u \rangle_{\mathcal{U}} - \langle u, \hat{\mathbf{b}} \rangle_{\mathcal{U}} - \frac{1}{2}\langle u, \widehat{\mathcal{C}}_W u \rangle_{\mathcal{U}} \\
&= \langle \sum_{i=1}^{n} \alpha_i' l(w_i, \cdot), u_{\boldsymbol{\alpha}} + u_{\perp} \rangle_{\mathcal{U}} - \langle u_{\boldsymbol{\alpha}} + u_{\perp}, \sum_{i=1}^{n} \alpha_i'' l(w_i, \cdot) \rangle_{\mathcal{U}} \\
&\quad - \frac{1}{2}\langle u_{\boldsymbol{\alpha}} + u_{\perp}, \widehat{\mathcal{C}}_W(u_{\boldsymbol{\alpha}} + u_{\perp}) \rangle_{\mathcal{U}} \\
&= \langle \sum_{i=1}^{n} \alpha_i' l(w_i, \cdot), u_{\boldsymbol{\alpha}} \rangle_{\mathcal{U}} - \langle u_{\boldsymbol{\alpha}}, \sum_{i=1}^{n} \alpha_i'' l(w_i, \cdot) \rangle_{\mathcal{U}} - \frac{1}{2}\langle u_{\boldsymbol{\alpha}}, \widehat{\mathcal{C}}_W u_{\boldsymbol{\alpha}} \rangle_{\mathcal{U}}.
\end{aligned}$$

The first equality follows from $\langle f, \widehat{\mathcal{C}}_{XW} u \rangle_{\mathcal{F}} = \langle \widehat{\mathcal{C}}_{WX} f, u \rangle_{\mathcal{U}}$ as $\widehat{\mathcal{C}}_{XW}$ is an adjoint operator of $\widehat{\mathcal{C}}_{WX}$. Since the choice of $f$ is arbitrary, the maximizer of $\widehat{\Psi}(f, u)$ with respect to $u$ also lives in the subspace $\mathcal{U}_{\boldsymbol{\alpha}}$.

Consequently, $\widehat{\Psi}(f, u) = \widehat{\Psi}(f_{\boldsymbol{\beta}}, u_{\boldsymbol{\alpha}})$ for some $\boldsymbol{\beta} \in \mathbb{R}^n$ and $\boldsymbol{\alpha} \in \mathbb{R}^n$. This completes the proof. $\square$

### E.3 Proof of Proposition 6

**Proposition 6.** *Given an i.i.d. sample $(x_i, y_i, z_i)_{i=1}^n$ from $\mathbb{P}(X, Y, Z)$, let $\mathbf{K} := \Phi^\top \Phi$ and $\mathbf{L} := \Upsilon^\top \Upsilon$ be the Gram matrices such that $\mathbf{K}_{ij} = k(x_i, x_j)$ and $\mathbf{L}_{ij} = l(w_i, w_j)$ where $w_i := (y_i, z_i)$. Then, $\hat{f} = \Phi \boldsymbol{\beta}$ where $\boldsymbol{\beta} = (\mathbf{MK} + n\lambda_2\mathbf{K})^{-1}\mathbf{My}$ and $\mathbf{M} := \mathbf{K}(\mathbf{L} + n\lambda_1 I)^{-1}\mathbf{L}$.*

*Proof.* It follows from (10) that the structural function $f^* \in \mathcal{F}$ satisfies

$$(\mathcal{C}_{XW}(\mathcal{C}_W + \lambda_1\mathcal{I})^{-1}\mathcal{C}_{WX} + \lambda_2\mathcal{I})f^* = \mathcal{C}_{XW}(\mathcal{C}_W + \lambda_1\mathcal{I})^{-1}\mathbf{b}.$$

Replacing the population quantities with the empirical counterparts $\widehat{\mathcal{C}}_{XW} = \frac{1}{n}\Phi\Upsilon^\top, \widehat{\mathcal{C}}_W = \frac{1}{n}\Upsilon\Upsilon^\top$, and $\hat{\mathbf{b}} = \frac{1}{n}\Upsilon\mathbf{y}$ yields

$$(\Phi\Upsilon^\top(\Upsilon\Upsilon^\top + n\lambda_1\mathcal{I})^{-1}\Upsilon\Phi^\top + n\lambda_2\mathcal{I})f^* = \Phi\Upsilon^\top(\Upsilon\Upsilon^\top + n\lambda_1\mathcal{I})^{-1}\Upsilon\mathbf{y}.$$

Using the identity $\Upsilon^\top(\Upsilon\Upsilon^\top + n\lambda_1\mathcal{I})^{-1} = (\Upsilon^\top\Upsilon + n\lambda_1 I)^{-1}\Upsilon^\top$, the above equation can be rewritten as

$$(\Phi(\Upsilon^\top\Upsilon + n\lambda_1 I)^{-1}\Upsilon^\top\Upsilon\Phi^\top + n\lambda_2\mathcal{I})f^* = \Phi(\Upsilon^\top\Upsilon + n\lambda_1 I)^{-1}\Upsilon^\top\Upsilon\mathbf{y}$$
$$(\Phi(\mathbf{L} + n\lambda_1 I)^{-1}\mathbf{L}\Phi^\top + n\lambda_2\mathcal{I})f^* = \Phi(\mathbf{L} + n\lambda_1 I)^{-1}\mathbf{L}\mathbf{y}.$$

By Lemma 5, $f^* = \Phi\boldsymbol{\beta}$ for some $\boldsymbol{\beta} \in \mathbb{R}^n$. Substituting this back into the equation above yields

$$\Phi(\mathbf{L} + n\lambda_1 I)^{-1}\mathbf{L}\Phi^\top\Phi\boldsymbol{\beta} + n\lambda_2\Phi\boldsymbol{\beta} = \Phi(\mathbf{L} + n\lambda_1 I)^{-1}\mathbf{L}\mathbf{y}$$
$$\Phi(\mathbf{L} + n\lambda_1 I)^{-1}\mathbf{L}\mathbf{K}\boldsymbol{\beta} + n\lambda_2\Phi\boldsymbol{\beta} = \Phi(\mathbf{L} + n\lambda_1 I)^{-1}\mathbf{L}\mathbf{y}.$$

Multiplying both sides of the equation by $\Phi^\top$ gives

$$\Phi^\top\Phi(\mathbf{L} + n\lambda_1 I)^{-1}\mathbf{L}\mathbf{K}\boldsymbol{\beta} + n\lambda_2\Phi^\top\Phi\boldsymbol{\beta} = \Phi^\top\Phi(\mathbf{L} + n\lambda_1 I)^{-1}\mathbf{L}\mathbf{y}$$
$$\mathbf{K}(\mathbf{L} + n\lambda_1 I)^{-1}\mathbf{L}\mathbf{K}\boldsymbol{\beta} + n\lambda_2\mathbf{K}\boldsymbol{\beta} = \mathbf{K}(\mathbf{L} + n\lambda_1 I)^{-1}\mathbf{L}\mathbf{y}$$
$$(\mathbf{K}(\mathbf{L} + n\lambda_1 I)^{-1}\mathbf{L}\mathbf{K} + n\lambda_2\mathbf{K})\boldsymbol{\beta} = \mathbf{K}(\mathbf{L} + n\lambda_1 I)^{-1}\mathbf{L}\mathbf{y}.$$

Setting $\mathbf{M} = \mathbf{K}(\mathbf{L} + n\lambda_1 I)^{-1}\mathbf{L}$ yields the result. $\square$

### E.4 Proof of Theorem 7

**Theorem 7.** *Let $\hat{f}_\lambda$ be an empirical estimator of $f^*$ obtained from Proposition 6 with the regularization parameters $\lambda := (\lambda_1, \lambda_2)$. Assume that $\mathcal{C}_W^{-1}$ and $(\mathcal{C}_{XW}\mathcal{C}_W^{-1}\mathcal{C}_{WX})^{-1}$ exist and the operator norm of the inverse are bounded. Then, for sufficiently slow decay of regularization parameters $\lambda_1$ and $\lambda_2$, $\hat{f}_\lambda$ is a consistent estimator of $f^*$ in RKHS norm, i.e., $\|\hat{f}_\lambda - f^*\|_{\mathcal{F}} \to 0$ as $n \to \infty$.*

For ease of understanding, we will use the following notation throughout the proof:

$$
\begin{aligned}
\mathcal{R} &:= \mathcal{C}_{XW}\mathcal{C}_W^{-1}\mathcal{C}_{WX} & \widehat{\mathcal{R}} &:= \widehat{\mathcal{C}}_{XW}\widehat{\mathcal{C}}_W^{-1}\widehat{\mathcal{C}}_{WX} \\
\mathcal{R}_{\lambda_1} &:= \mathcal{C}_{XW}(\mathcal{C}_W + \lambda_1\mathcal{I})^{-1}\mathcal{C}_{WX} & \widehat{\mathcal{R}}_{\lambda_1} &:= \widehat{\mathcal{C}}_{XW}(\widehat{\mathcal{C}}_W + \lambda_1\mathcal{I})^{-1}\widehat{\mathcal{C}}_{WX} \\
\mathcal{C}_{\lambda_1} &:= \mathcal{C}_W + \lambda_1\mathcal{I} & \widehat{\mathcal{C}}_{\lambda_1} &:= \widehat{\mathcal{C}}_W + \lambda_1\mathcal{I}.
\end{aligned}
$$

The following identity will be used heavily in our proof:

$$(B^{-1} - A^{-1}) = B^{-1}(A - B)A^{-1}. \tag{14}$$

*Proof.* First, it follows from the assumption that $\|\mathcal{C}_W^{-1}\|_{op} \leq \delta_1^{-1}$ and $\|(\mathcal{C}_{XW}\mathcal{C}_W^{-1}\mathcal{C}_{WX}^\top)^{-1}\|_{op} \leq \delta_2^{-1}$ for some $\delta_1, \delta_2 > 0$. Moreover, we can write the empirical estimate as

$$\hat{f}_\lambda = (\widehat{\mathcal{R}}_{\lambda_1} + \lambda_2\mathcal{I})^{-1}\widehat{\mathcal{C}}_{XW}\widehat{\mathcal{C}}_{\lambda_1}^{-1}\hat{\mathbf{b}}.$$

Similarly, under our assumption, the true population function can be expressed as

$$f^* = \mathcal{R}^{-1}\mathcal{C}_{XW}\mathcal{C}_W^{-1}\mathbf{b}.$$

The goal is then to bound the difference of $\hat{f}_\lambda$ and $f^*$ in RKHS norm, *i.e.*,

$$
\begin{aligned}
\left\|\hat{f}_\lambda - f^*\right\|_{\mathcal{F}} &= \left\|(\widehat{\mathcal{R}}_{\lambda_1} + \lambda_2 \mathcal{I})^{-1}\widehat{\mathcal{C}}_{XW}\widehat{\mathcal{C}}_{\lambda_1}^{-1}\hat{\mathbf{b}} - \mathcal{R}^{-1}\mathcal{C}_{XW}\mathcal{C}_W^{-1}\mathbf{b}\right\|_{\mathcal{F}} \\
&\leq \left\|(\widehat{\mathcal{R}}_{\lambda_1} + \lambda_2 \mathcal{I})^{-1}\widehat{\mathcal{C}}_{XW}\widehat{\mathcal{C}}_{\lambda_1}^{-1}\hat{\mathbf{b}} - (\mathcal{R} + \lambda_2 I)^{-1}\mathcal{C}_{XW}\mathcal{C}_{\lambda_1}^{-1}\mathbf{b}\right\|_{\mathcal{F}} \\
&\quad + \left\|(\mathcal{R}_{\lambda_1} + \lambda_2 \mathcal{I})^{-1}\mathcal{C}_{XW}\mathcal{C}_{\lambda_1}^{-1}\mathbf{b} - \mathcal{R}^{-1}\mathcal{C}_{XW}\mathcal{C}_W^{-1}\mathbf{b}\right\|_{\mathcal{F}} \\
&=: \; T_1 + T_2.
\end{aligned}
\tag{15}
$$

**Bounding $T_2$:** Let us first consider the second term $T_2$ in (15):

$$
\begin{aligned}
T_2 &:= \left\|(\mathcal{R}_{\lambda_1} + \lambda_2 \mathcal{I})^{-1}\mathcal{C}_{XW}\mathcal{C}_{\lambda_1}^{-1}\mathbf{b} - \mathcal{R}^{-1}\mathcal{C}_{XW}\mathcal{C}_W^{-1}\mathbf{b}\right\|_{\mathcal{F}} \\
&\leq \left\|(\mathcal{R}_{\lambda_1} + \lambda_2 \mathcal{I})^{-1}\mathcal{C}_{XW}\mathcal{C}_{\lambda_1}^{-1}\mathbf{b} - \mathcal{R}_{\lambda_1}^{-1}\mathcal{C}_{XW}\mathcal{C}_{\lambda_1}^{-1}\mathbf{b}\right\|_{\mathcal{F}} \\
&\quad + \left\|\mathcal{R}_{\lambda_1}^{-1}\mathcal{C}_{XW}\mathcal{C}_{\lambda_1}^{-1}\mathbf{b} - \mathcal{R}^{-1}\mathcal{C}_{XW}\mathcal{C}_W^{-1}\mathbf{b}\right\|_{\mathcal{F}} \\
&=: \; T_{21} + T_{22}.
\end{aligned}
\tag{16}
$$

**Bounding $T_{21}$:** Let us consider $T_{21}$ in (16) first.

$$
\begin{aligned}
T_{21} &= \left\|(\mathcal{R}_{\lambda_1} + \lambda_2 \mathcal{I})^{-1}\mathcal{C}_{XW}\mathcal{C}_{\lambda_1}^{-1}\mathbf{b} - \mathcal{R}_{\lambda_1}^{-1}\mathcal{C}_{XW}\mathcal{C}_{\lambda_1}^{-1}\mathbf{b}\right\|_{\mathcal{F}} \\
&= \lambda_2 \left\|(\mathcal{R}_{\lambda_1} + \lambda_2 \mathcal{I})^{-1}\mathcal{R}_{\lambda_1}^{-1}\mathcal{C}_{XW}\mathcal{C}_{\lambda_1}^{-1}\mathbf{b}\right\|_{\mathcal{F}} \\
&\leq \lambda_2 \left\|(\mathcal{R}_{\lambda_1} + \lambda_2 \mathcal{I})^{-1}\right\|_{op} \left\|\mathcal{R}_{\lambda_1}^{-1}\right\|_{op} \left\|\mathcal{C}_{XW}\mathcal{C}_{\lambda_1}^{-1}\mathbf{b}\right\|_{\mathcal{F}} \\
&\leq \lambda_2 \left\|(\mathcal{R}_{\lambda_1} + \lambda_2 \mathcal{I})^{-1}(\mathcal{C}_{XW}\mathcal{C}_{\lambda_1}^{-1}\mathcal{C}_{WX})\right\|_{op} \left\|\mathcal{R}_{\lambda_1}^{-1}\right\|_{op}^2 \left\|\mathcal{C}_{XW}\mathcal{C}_{\lambda_1}^{-1}\mathbf{b}\right\|_{\mathcal{F}} \\
&\leq \lambda_2 \left\|\mathcal{R}_{\lambda_1}^{-1}\right\|_{op}^2 \left\|\mathcal{C}_{XW}\mathcal{C}_{\lambda_1}^{-1}\mathbf{b}\right\|_{\mathcal{F}} \\
&\leq \frac{\lambda_2}{\delta_2^2}\left\|\mathcal{C}_{XW}\mathcal{C}_{\lambda_1}^{-1}\mathbf{b}\right\|_{\mathcal{F}} \leq \frac{\lambda_2}{\delta_1\delta_2^2}\left\|\mathcal{C}_{XW}\right\|_{op}\|\mathbf{b}\|_{\mathcal{F}}.
\end{aligned}
\tag{17}
$$

The second equality in (17) follows from the identity in (14). Hence, we have $T_{21} = \mathcal{O}(\lambda_2)$. From the above argument, it is also clear that there exists a positive constant $C$ such that $\max(\|\mathcal{C}_{XW}\mathcal{C}_W^{-1}\mathbf{b}\|_{\mathcal{F}}, \|\mathcal{C}_{XW}\mathcal{C}_{\lambda_1}^{-1}\mathbf{b}\|_{\mathcal{F}}) \leq C$.

**Bounding $T_{22}$:** Let us now consider the term $T_{22}$ in (16).

$$
\begin{aligned}
T_{22} &= \left\|\mathcal{R}_{\lambda_1}^{-1}\mathcal{C}_{XW}\mathcal{C}_{\lambda_1}^{-1}\mathbf{b} - \mathcal{R}^{-1}\mathcal{C}_{XW}\mathcal{C}_W^{-1}\mathbf{b}\right\|_{\mathcal{F}} \\
&\leq \left\|\mathcal{R}_{\lambda_1}^{-1}\mathcal{C}_{XW}\mathcal{C}_{\lambda_1}^{-1}\mathbf{b} - \mathcal{R}^{-1}\mathcal{C}_{XW}\mathcal{C}_{\lambda_1}^{-1}\mathbf{b}\right\|_{\mathcal{F}} + \left\|\mathcal{R}^{-1}\mathcal{C}_{XW}\mathcal{C}_{\lambda_1}^{-1}\mathbf{b} - \mathcal{R}^{-1}\mathcal{C}_{XW}\mathcal{C}_W^{-1}\mathbf{b}\right\|_{\mathcal{F}} \\
&= \left\|(\mathcal{R}_{\lambda_1}^{-1} - \mathcal{R}^{-1})\mathcal{C}_{XW}\mathcal{C}_{\lambda_1}^{-1}\mathbf{b}\right\|_{\mathcal{F}} + \left\|\mathcal{R}^{-1}(\mathcal{C}_{XW}\mathcal{C}_{\lambda_1}^{-1}\mathbf{b} - \mathcal{C}_{XW}\mathcal{C}_W^{-1}\mathbf{b})\right\|_{\mathcal{F}} \\
&\leq C\left\|\mathcal{R}_{\lambda_1}^{-1}\right\|_{op}\left\|\mathcal{R}^{-1}\right\|_{op}\left\|\mathcal{R}_{\lambda_1} - \mathcal{R}\right\|_{op} + \left\|\mathcal{R}^{-1}\right\|_{op}\left\|\mathcal{C}_{XW}(\mathcal{C}_{\lambda_1}^{-1} - \mathcal{C}_W^{-1})\mathbf{b}\right\|_{\mathcal{F}} \\
&\leq C\left\|\mathcal{R}_{\lambda_1}^{-1}\right\|_{op}\left\|\mathcal{R}^{-1}\right\|_{op}\left\|\mathcal{C}_{XW}(\mathcal{C}_{\lambda_1}^{-1} - \mathcal{C}_W^{-1})\mathcal{C}_{WX}\right\|_{op} \\
&\quad + \left\|\mathcal{R}^{-1}\right\|_{op}\left\|\mathcal{C}_{XW}(\mathcal{C}_{\lambda_1}^{-1} - \mathcal{C}_W^{-1})\mathbf{b}\right\|_{\mathcal{F}}.
\end{aligned}
\tag{18}
$$

Further, we have

$$
T_{22} \leq \frac{\lambda C}{\delta_2}\left\|\mathcal{R}_{\lambda_1}^{-1}\right\|_{op}\left\|\mathcal{C}_{XW}\mathcal{C}_{\lambda_1}^{-1}\mathcal{C}_W^{-1}\mathcal{C}_{WX}\right\|_{op} + \left\|\mathcal{R}^{-1}\right\|_{op}\left\|\mathcal{C}_{XW}(\mathcal{C}_{\lambda_1}^{-1} - \mathcal{C}_W^{-1})\mathbf{b}\right\|_{\mathcal{F}}.
\tag{19}
$$

Let us consider now the following term in the above inequality:

$$
\begin{aligned}
\left\|\mathcal{R}_{\lambda_1}^{-1}\right\|_{op} &\leq \left\|\mathcal{R}^{-1}\right\|_{op} + \left\|\mathcal{R}_{\lambda_1}^{-1} - \mathcal{R}^{-1}\right\|_{op} \\
&\leq \frac{1}{\delta_2} + \left\|\mathcal{R}_{\lambda_1}^{-1} - \mathcal{R}^{-1}\right\|_{op} \\
&\leq \frac{1}{\delta_2} + \left\|\mathcal{R}_{\lambda_1}^{-1}\right\|_{op}\left\|\mathcal{R}^{-1}\right\|_{op}\left\|\mathcal{R}_{\lambda_1} - \mathcal{R}\right\|_{op}
\end{aligned}
$$

$$= \frac{1}{\delta_2} + \left\|\mathcal{R}_{\lambda_1}^{-1}\right\|_{op} \left\|\mathcal{R}^{-1}\right\|_{op} \left\|\mathcal{C}_{XW}(\mathcal{C}_{\lambda_1}^{-1} - \mathcal{C}_W^{-1})\mathcal{C}_{WX}\right\|_{op}$$

$$= \frac{1}{\delta_2} + \lambda_1 \left\|\mathcal{R}_{\lambda_1}^{-1}\right\|_{op} \left\|\mathcal{R}^{-1}\right\|_{op} \left\|\mathcal{C}_{XW}\mathcal{C}_{\lambda_1}^{-1}\mathcal{C}_W^{-1}\mathcal{C}_{WX}\right\|_{op}.$$

Now, $\|\mathcal{C}_{XW}\mathcal{C}_{\lambda_1}^{-1}\mathcal{C}_W^{-1}\mathcal{C}_{WX}\|_{op} \leq \tilde{c}/\delta_1^2$ and similarly $\|\mathcal{C}_{XW}(\mathcal{C}_{\lambda_1}^{-1} - \mathcal{C}_W^{-1})\mathbf{b}\|_{\mathcal{F}} \leq \hat{c}/\delta_1^2$ for some positive real numbers $\hat{c}$ and $\tilde{c}$. Hence,

$$\left\|\mathcal{R}_{\lambda_1}^{-1}\right\|_{op} \leq \frac{1}{\delta_2} + \frac{\lambda_1 \tilde{c}}{\delta_1^2 \delta_2} \left\|\mathcal{R}_{\lambda_1}^{-1}\right\|_{op} \quad \Rightarrow \quad \left\|\mathcal{R}_{\lambda_1}^{-1}\right\|_{op} \leq \frac{1/\delta_2}{1 - \frac{\lambda_1 \tilde{c}}{\delta_1^2 \delta_2}}.$$

Hence, if $\lambda_1 \to 0$ sufficiently fast, then

$$T_{22} \leq \frac{\lambda_1 \tilde{C}}{\delta_1^2 \delta_2} \tag{20}$$

for a positive real number $\tilde{C}$. This implies $T_{22} = \mathcal{O}(\lambda_1)$.

**Bounding $T_1$:**  We now consider the first term in (15).

$$
\begin{aligned}
T_1 &= \left\|(\widehat{\mathcal{R}}_{\lambda_1} + \lambda_2\mathcal{I})^{-1}\widehat{\mathcal{C}}_{XW}\widehat{\mathcal{C}}_{\lambda_1}^{-1}\hat{\mathbf{b}} - (\mathcal{R}_{\lambda_1} + \lambda_2\mathcal{I})^{-1}\mathcal{C}_{XW}\mathcal{C}_{\lambda_1}^{-1}\mathbf{b}\right\|_{\mathcal{F}} \\
&\leq \left\|(\widehat{\mathcal{R}}_{\lambda_1} + \lambda_2\mathcal{I})^{-1}\widehat{\mathcal{C}}_{XW}\widehat{\mathcal{C}}_{\lambda_1}^{-1}\hat{\mathbf{b}} - (\mathcal{R}_{\lambda_1} + \lambda_2\mathcal{I})^{-1}\widehat{\mathcal{C}}_{XW}\widehat{\mathcal{C}}_{\lambda_1}^{-1}\hat{\mathbf{b}}\right\|_{\mathcal{F}} \\
&\quad + \left\|(\mathcal{R}_{\lambda_1} + \lambda_2\mathcal{I})^{-1}\widehat{\mathcal{C}}_{XW}\widehat{\mathcal{C}}_{\lambda_1}^{-1}\hat{\mathbf{b}} - (\mathcal{R}_{\lambda_1} + \lambda_2\mathcal{I})^{-1}\mathcal{C}_{XW}\mathcal{C}_{\lambda_1}^{-1}\mathbf{b}\right\|_{\mathcal{F}} \\
&=: T_{11} + T_{12}. \tag{21}
\end{aligned}
$$

**Bounding $T_{11}$:**  Consider the first term in (21):

$$
\begin{aligned}
T_{11} &= \left\|(\widehat{\mathcal{R}}_{\lambda_1} + \lambda_2\mathcal{I})^{-1}\widehat{\mathcal{C}}_{XW}\widehat{\mathcal{C}}_{\lambda_1}^{-1}\hat{\mathbf{b}} - (\mathcal{R}_{\lambda_1} + \lambda_2\mathcal{I})^{-1}\widehat{\mathcal{C}}_{XW}\widehat{\mathcal{C}}_{\lambda_1}^{-1}\hat{\mathbf{b}}\right\|_{\mathcal{F}} \\
&\leq \left\|\widehat{\mathcal{C}}_{XW}\widehat{\mathcal{C}}_{\lambda_1}^{-1}\hat{\mathbf{b}}\right\|_{\mathcal{F}} \left\|(\widehat{\mathcal{R}}_{\lambda_1} + \lambda_2\mathcal{I})^{-1}\right\|_{op} \left\|(\mathcal{R}_{\lambda_1} + \lambda_2\mathcal{I})^{-1}\right\|_{op} \left\|\widehat{\mathcal{R}}_{\lambda_1} - \mathcal{R}_{\lambda_1}\right\|_{op} \tag{22} \\
&\leq \frac{C}{\lambda_1 \lambda_2^2} \left\|\widehat{\mathcal{R}}_{\lambda_1} - \mathcal{R}_{\lambda_1}\right\|_{op}
\end{aligned}
$$

for some positive constant $C$. Next, we have

$$
\begin{aligned}
&\left\|\widehat{\mathcal{C}}_{XW}\widehat{\mathcal{C}}_{\lambda_1}^{-1}\widehat{\mathcal{C}}_{WX} - \mathcal{C}_{XW}\mathcal{C}_{\lambda_1}^{-1}\mathcal{C}_{WX}\right\|_{op} \\
&\leq \left\|\widehat{\mathcal{C}}_{XW}\widehat{\mathcal{C}}_{\lambda_1}^{-1}\widehat{\mathcal{C}}_{WX} - \widehat{\mathcal{C}}_{XW}\mathcal{C}_{\lambda_1}^{-1}\widehat{\mathcal{C}}_{WX}\right\|_{op} + \left\|\widehat{\mathcal{C}}_{XW}\mathcal{C}_{\lambda_1}^{-1}\widehat{\mathcal{C}}_{WX} - \mathcal{C}_{XW}\mathcal{C}_{\lambda_1}^{-1}\mathcal{C}_{WX}\right\|_{op} \\
&\leq \left\|\widehat{\mathcal{C}}_{XW}(\widehat{\mathcal{C}}_{\lambda_1}^{-1} - \mathcal{C}_{\lambda_1}^{-1})\widehat{\mathcal{C}}_{WX}\right\|_{op} + \left\|\widehat{\mathcal{C}}_{XW}\mathcal{C}_{\lambda_1}^{-1}\widehat{\mathcal{C}}_{WX} - \mathcal{C}_{XW}\mathcal{C}_{\lambda_1}^{-1}\widehat{\mathcal{C}}_{WX}\right\|_{op} \\
&\quad + \left\|\mathcal{C}_{XW}\mathcal{C}_{\lambda_1}^{-1}\widehat{\mathcal{C}}_{WX} - \mathcal{C}_{XW}\mathcal{C}_{\lambda_1}^{-1}\mathcal{C}_{WX}\right\|_{op} \tag{23} \\
&\leq \left\|\widehat{\mathcal{C}}_{XW}\widehat{\mathcal{C}}_{\lambda_1}^{-1}(\widehat{\mathcal{C}}_W - \mathcal{C}_W)\mathcal{C}_{\lambda_1}^{-1}\widehat{\mathcal{C}}_{WX}\right\|_{op} + \left\|(\widehat{\mathcal{C}}_{XW} - \mathcal{C}_{XW})\mathcal{C}_{\lambda_1}^{-1}\widehat{\mathcal{C}}_{WX}\right\|_{op} \\
&\quad + \left\|\mathcal{C}_{XW}\mathcal{C}_{\lambda_1}^{-1}(\widehat{\mathcal{C}}_{WX} - \mathcal{C}_{WX})\right\|_{op}.
\end{aligned}
$$

From the $\sqrt{n}$-consistency of covariance and cross-covariance operators [47, 48], we have

$$\left\|\widehat{\mathcal{C}}_{XW}\widehat{\mathcal{C}}_{\lambda_1}^{-1}\widehat{\mathcal{C}}_{WX} - \mathcal{C}_{XW}\mathcal{C}_{\lambda_1}^{-1}\mathcal{C}_{WX}\right\|_{op} = \mathcal{O}\left(\frac{1}{\lambda_1\sqrt{n}}\right).$$

Hence, if $(\lambda_1\lambda_2)^2$ converges to zero slower than $1/\sqrt{n}$, then $T_{11}$ converges to zero asymptotically.

**Bounding $T_{12}$:**    Let us now consider the second term in (21):

$$
\begin{aligned}
T_{12} &= \left\| (\mathcal{R}_{\lambda_1} + \lambda_2 \mathcal{I})^{-1} \widehat{\mathcal{C}}_{XW} \widehat{\mathcal{C}}_{\lambda_1}^{-1} \hat{\mathbf{b}} - (\mathcal{R}_{\lambda_1} + \lambda_2 \mathcal{I})^{-1} \mathcal{C}_{XW} \mathcal{C}_{\lambda_1}^{-1} \mathbf{b} \right\|_{\mathcal{F}} \\
&\le \left\| (\mathcal{R}_{\lambda_1} + \lambda_2 \mathcal{I})^{-1} \right\|_{op} \left\| \widehat{\mathcal{C}}_{XW} \widehat{\mathcal{C}}_{\lambda_1}^{-1} \hat{\mathbf{b}} - \mathcal{C}_{XW} \mathcal{C}_{\lambda_1}^{-1} \mathbf{b} \right\|_{\mathcal{F}} \\
&\le \frac{1}{\lambda_2} \left\| \widehat{\mathcal{C}}_{XW} \widehat{\mathcal{C}}_{\lambda_1}^{-1} \hat{\mathbf{b}} - \mathcal{C}_{XW} \mathcal{C}_{\lambda_1}^{-1} \mathbf{b} \right\|_{\mathcal{F}} \\
&\le \frac{1}{\lambda_2} \left[ \left\| \widehat{\mathcal{C}}_{XW} \widehat{\mathcal{C}}_{\lambda_1}^{-1} \hat{\mathbf{b}} - \mathcal{C}_{XW} \widehat{\mathcal{C}}_{\lambda_1}^{-1} \hat{\mathbf{b}} \right\|_{\mathcal{F}} + \left\| \mathcal{C}_{XW} \widehat{\mathcal{C}}_{\lambda_1}^{-1} \hat{\mathbf{b}} - \mathcal{C}_{XW} \mathcal{C}_{\lambda_1}^{-1} \hat{\mathbf{b}} \right\|_{\mathcal{F}} \right. \\
&\qquad \left. + \left\| \mathcal{C}_{XW} \mathcal{C}_{\lambda_1}^{-1} \hat{\mathbf{b}} - \mathcal{C}_{XW} \mathcal{C}_{\lambda_1}^{-1} \mathbf{b} \right\|_{\mathcal{F}} \right] \\
&\le \frac{1}{\lambda_2} \left[ \left\| (\widehat{\mathcal{C}}_{XW} - \mathcal{C}_{XW}) \widehat{\mathcal{C}}_{\lambda_1}^{-1} \hat{\mathbf{b}} \right\|_{\mathcal{F}} + \left\| \mathcal{C}_{XW} (\widehat{\mathcal{C}}_{\lambda_1}^{-1} - \mathcal{C}_{\lambda_1}^{-1}) \hat{\mathbf{b}} \right\|_{\mathcal{F}} \right. \\
&\qquad \left. + \left\| \mathcal{C}_{XW} \mathcal{C}_{\lambda_1}^{-1} \hat{\mathbf{b}} - \mathcal{C}_{XW} \mathcal{C}_{\lambda_1}^{-1} \mathbf{b} \right\|_{\mathcal{G}} \right].
\end{aligned}
\tag{24}
$$

By the $\sqrt{n}$-consistency of mean element and covariance operator in RKHS [47, 48], we have that $T_{12} = \mathcal{O}\left( \frac{1}{\lambda_1 \lambda_2 \sqrt{n}} \right)$. Moreover, it follows from what we have shown so far that

$$
\begin{aligned}
&\left\| (\widehat{\mathcal{C}}_{XW} \widehat{\mathcal{C}}_{\lambda_1}^{-1} \widehat{\mathcal{C}}_{WX} + \lambda_2 \mathcal{I})^{-1} \widehat{\mathcal{C}}_{XW} \widehat{\mathcal{C}}_{\lambda_1}^{-1} \hat{\mathbf{b}} - (\mathcal{C}_{XW} \mathcal{C}_W^{-1} \mathcal{C}_{WX})^{-1} \mathcal{C}_{XW} \mathcal{C}_W^{-1} \mathbf{b} \right\|_{\mathcal{F}} \\
&\qquad \le T_1 + T_2 \le T_{11} + T_{12} + T_{13} + T_{14}.
\end{aligned}
\tag{25}
$$

Hence, if $\lambda_1$ and $\lambda_2$ converge to zero with the sample size $n$ such that $\frac{1}{\lambda_1^2 \lambda_2^2 \sqrt{n}}$ also converges to zero, then $\| \hat{f}_\lambda - f^* \|_{\mathcal{F}} \to 0$. That is, our estimator is consistent in RKHS norm. $\qquad\square$