[Reviews · NeurIPS 2020]

Review 1

Summary and Contributions: The paper proposed a novel framework for instrumental variable regression, called DualIV, that directly estimates the structural causal function. The derivation is rooted in dual optimization and it overcomes a longstanding challenge that prevents two-stage regression in estimating causal inference via instrumental variables. ===== I have read the authors' feedback carefully and I do appreciate the author's reply on the scalability concern. They have claimed that it is not adequately addressed in this paper and will include more details in the camera-ready version. I am looking forward to their updated one. The merit of this paper and the beautiful insight will not be overshadowed by this issue, it would be better if they can discuss or solve this problem at least to some extent.

Strengths: I like the idea of this paper. The paper is of high quality and provides theoretical proofs and extensive experimental results. Duality is a very nice, clean, and useful approach to combining causality and stochastic programming. The paper has the potential of conveying the message of causality into the wider machine learning community and thereby trigger other ideas in this area.

Weaknesses: The general perception is that kernel methods are not scalable. I am a little bit worry about the performance of dualIV on the large-scale data, and I am willing to see more experimental results (e.g. the sigmoid function f in Kernel IV paper).

Correctness: The claims and method are correct and the experimental results are promising.

Clarity: The paper is very clearly written and well organized, a pleasure to read. The assumptions are clearly stated.

Relation to Prior Work: They clearly clarified the main contribution of the work relative to these previous works.

Reproducibility: Yes

Additional Feedback:


Review 2

Summary and Contributions: This paper studies the regression problem, fittiing the function f with Y~E(f(X)|Z) via its dual formulation. This applies on least squares problem with the unsupervised approach. The paper proposes an algorithm based on this setting equipped with parameter selection, theoretical analysis and experiments to justify the algorithm. ======================= I have seen the feedback and I'm convinced by the reply to my doubts. I think this paper is correct now, but I'd like to see those points are explained more clearly in the final version. I think it's okay to accept it as a main conf paper, but since the technical difficulty is fairly low and the method to some extent origins from what people have known or used, I'd keep my score of 6.

Strengths: Concrete analysis of dual formulation. Clear explanation of the roadmap towards the idea of the working algorithm.

Weaknesses: I do not see major weaknesses, some minor discussions below.

Correctness: I believe the theory is correct and the experiments are reasonable.

Clarity: Yes the ideas are conveyed clearly.

Relation to Prior Work: Sorry I'm not quite familliar with the area. I appreciate the list of related literature and discussions in the first two sections.

Reproducibility: Yes

Additional Feedback: 1. Line 54, could the authors explain more about do(X=x)? And I'd like to see it emphasized later in section 3&4, what it, as well as correlated noise, means and obstacles for the formulation of the problem. 2. I don't quite see the paragraph Line 130 "cumbersome to solve (4)...". If you can do E_{XYZ}, does it mean you know p(X,Y,Z), and why is E_{X|Z} hard? Is it hard to formulate the RKHS?


Review 3

Summary and Contributions: This paper addresses the problem of instrumental variable regression. It introduces a dual formulation for the corresponding learning problem, which boils down to be a convex-concave saddle-point problem. Restricting the hypothesis set to an RKHS, the authors present a closed-form solution to the minimax problem, leading to an easy implementation of the desired estimator.

Strengths: The contribution is algorithmic since the paper introduces a dual formulation and a closed-form solution to a known problem (intrsumental variable regression). I find it very interesting and definitely relevant for the NeurIPS community. The paper is well written and the proposed approach is clearly explained. The work is complete since the method is supported by theoretical and numerical evidences.

Weaknesses: I only have minor comments: - Page 2, Line 54: more explanation is needed about conditionning to do(X=x). - Page 6, Line 233: clarify dimensions of objects. - Page 7, Line 243: \top means inner product in RKHS F?

Correctness: Everything seems correct.

Clarity: The paper is clear.

Relation to Prior Work: Relation to prior work is adequately addressed.

Reproducibility: Yes

Additional Feedback: I thank the authors for addressing my comments in their rebuttal.

[Author Response · NeurIPS 2020]

We thank all reviewers for the positive reviews. Overall, there is a consensus among the reviewers that our work is
deemed appropriate for publication at NeurIPS.

• **R2** advocated that "*the paper has the potential of conveying the message of causality into the wider machine learning*
*community and thereby trigger other ideas in this area.*"
• **R3** highlighted the strengths of the paper as "*[c]oncrete analysis of dual formulation [and] [c]lear explanation of the*
*roadmap towards the idea of the working algorithm.*"
• **R4** wrote that "*[ ... ] I find [the contribution] very interesting and definitely relevant for the NeurIPS community.*"

In the camera-ready version, we will take all comments into consideration. Below we respond to major concerns.

**The interpretation of the do$(\cdot)$ operator (R3,R4).**    This concept is at the core of our paper, so we thank the reviewers
for raising this point, which should have been made clearer in the initial submission. In our camera-ready version, we
will provide additional discussion around this concept.

In Line 54 of our submission, $\text{do}(X = x)$ denotes a mathematical operator which simulates physical interventions by
setting the value of $X$ to $x$, while keeping the rest of the model unchanged [Pearl, 2009, Sec. 3.2.1]. In this work, we
aim to estimate $\mathbb{E}[Y \mid \text{do}(X = x)]$ which is a conditional expectation computed w.r.t. the *interventional* distribution
$P(Y \mid \text{do}(X = x))$. We can estimate $P(Y \mid \text{do}(X = x))$ if it is possible to manipulate $X$ and then observe the resulting
outcome $Y$. In Figure 1, for example, one may assign different levels of education to people and then observe the
resulting levels of income when they enter the labor market. Unfortunately, such experiment is not always possible and
we only have access to an *observational* distribution $P(Y \mid X = x)$, which can be different from $P(Y \mid \text{do}(X = x))$.
In this example, the discrepancy results from the unobserved socioeconomic status, as illustrated in Figure 1.

**Scalability of DualIV (R2).**    The scalability is an important aspect that was not adequately addressed in our submission.
We thank the reviewer for pointing it out. We will discuss it in more detail in our camera-ready version.

To improve scalability of our method, we can employ the stochastic gradient descent (SGD) based algorithms similar to
those proposed in Dai et al. [2017, Algorithm 1] (also cited in our submission) to solve the dual formulation, i.e., Eq.
(7) in the submission. Based on SGD, other models such as deep neural networks can also be used to parametrize the
function classes $\mathcal{F}$ and $\mathcal{U}$ in Eq. (7). Furthermore, we can improve the scalability of the kernelized DualIV algorithm
(i.e., Algorithm 1 in our submission) by leveraging rich literature on large-scale kernel machines such as random Fourier
feature (RFF) and Nyström method; see, e.g., Yang et al. [2012] and references therein.

**Responses to reviewers' remaining questions.**

• (**R3**) In Line 130, we stated that it is "cumbersome to solve (4) ..." because solving (4) requires *two-stage* estimation.
In the first stage, $\mathbb{E}_{X|Z}[\cdot]$ must be estimated, which is challenging on its own because we only observe a single
sample $x_i$ from $P(X|Z = z_i)$ for each value $z_i$. The first-stage estimate is then used in the second stage to estimate
$\mathbb{E}_{YZ}[\cdot]$ in Eq. (4). Our contribution is precisely to reformulate the problem such that one can solve the problem in a
single step by estimating $\mathbb{E}_{XYZ}[\cdot]$ directly, as we did in Eq. (6).
• (**R4**) Page 7, Line 243: The notation $^\top$ translates to the inner product in RKHS $\mathcal{F}$ and $\mathcal{U}$. With slight abuse of
notation, we define $\Phi := [\phi(x_1), \ldots, \phi(x_n)]$ and $\Upsilon := [\varphi(y_1, z_1), \ldots, \varphi(y_n, z_n)]$. Hence, we have

$$\Phi^\top \Phi = \begin{bmatrix} \langle \phi(x_1), \phi(x_1) \rangle_{\mathcal{F}} & \cdots & \langle \phi(x_n), \phi(x_1) \rangle_{\mathcal{F}} \\ \vdots & \ddots & \vdots \\ \langle \phi(x_1), \phi(x_n) \rangle_{\mathcal{F}} & \cdots & \langle \phi(x_n), \phi(x_n) \rangle_{\mathcal{F}} \end{bmatrix} = \begin{bmatrix} k(x_1, x_1) & \cdots & k(x_n, x_1) \\ \vdots & \ddots & \vdots \\ k(x_1, x_n) & \cdots & k(x_n, x_n) \end{bmatrix} = \mathbf{K}.$$

The matrix $\Upsilon^\top \Upsilon$ is defined similarly. We will clarify this notation in our camera-ready version.

# References

B. Dai, N. He, Y. Pan, B. Boots, and L. Song.  Learning from Conditional Distributions via Dual Embeddings.
In *Proceedings of the 20th International Conference on Artificial Intelligence and Statistics*, volume 54, pages
1458–1467. PMLR, 2017.

J. Pearl. Causal inference in statistics: An overview. *Statistics Surveys*, 3:96–146, 2009.

T. Yang, Y.-f. Li, M. Mahdavi, R. Jin, and Z.-H. Zhou. Nyström method vs random fourier features: A theoretical and
empirical comparison. In *Advances in Neural Information Processing Systems 25*, pages 476–484. 2012.


[Meta-Review · NeurIPS 2020]

The paper studies the problem instrumental variable regression via its dual formulation. By restricting the hypothesis set to an RKHS, the authors can derive a closed-form solution to the corresponding minimax problem. The reviewers found both the theoretical results (especially the formulation in terms of the dual program) as well as the experimental results satisfying. Instrumental variables are important in causal inference and as such this paper could be of interest to the broader NeurIPS audience.